# LLM Watermark Evasion via Bias Inversion

## Abstract

Watermarking for large language models (LLMs) embeds a statistical signal during generation to enable detection of model-produced text. While watermarking has proven effective in benign settings, its robustness under adversarial evasion remains contested. To advance a rigorous understanding and evaluation of such vulnerabilities, we propose the *Bias-Inversion Rewriting Attack* (BIRA), which is theoretically motivated and model-agnostic. BIRA weakens the watermark signal by suppressing the logits of likely watermarked tokens during LLM-based rewriting, without any knowledge of the underlying watermarking scheme. Across recent watermarking methods, BIRA achieves over 99% evasion while preserving the semantic content of the original text. Beyond demonstrating an attack, our results reveal a systematic vulnerability, emphasizing the need for stress testing and robust defenses.

## 1 Introduction

The rapid advancement and proliferation of large language models (LLMs) (Minaee et al., 2024; Wang et al., 2024a) have intensified concerns about their misuse, ranging from the spread of misleading content (Monteith et al., 2024; Wang et al., 2024b; Papageorgiou et al., 2024) to threats to academic integrity, such as cheating (Stokel-Walker, 2022; Kamalov et al., 2023). To address these risks, watermarking has been proposed as a promising approach for detecting LLM-generated content (Aaronson & Kirchner, 2022; Kirchenbauer et al., 2024b). The core idea is to embed an imperceptible statistical signal into generated text, for example, by partitioning the vocabulary into "green" and "red" lists using a secret key, and biasing generation toward the green list. A detector then identifies LLM-generated text by checking for statistical overrepresentation of green tokens.

Recent work shows that watermarking is robust against common evasion strategies, such as text insertion, text substitution, and text deletion (Kirchenbauer et al., 2024b; Liu et al., 2024; Zhao et al., 2024; Lee et al., 2024; Lu et al., 2024). This robustness has drawn significant attention and spurred movement toward deployment. For instance, OpenAI has discussed adding watermarking to its products (Bartz & Hu, 2023), and U.S. policymakers have proposed legislation requiring watermarks for AI-generated content (Tong, 2024).

However, recent studies (Raffel et al., 2020; Cheng et al., 2025; Wu & Chandrasekaran, 2024; Chen et al., 2024; Jovanović et al., 2024; Diaa et al., 2024) have questioned the robustness of watermarking, noting that existing methods have not been sufficiently stress-tested and showing that watermarks can be evaded through sophisticated strategies. These approaches fall into two categories: query-based attacks (Wu & Chandrasekaran, 2024; Chen et al., 2024; Jovanović et al., 2024), which focus on query strategies to recover the green token set but require unrestricted access to the target model via repeated queries and often rely on knowledge of the watermarking scheme, and query-free attacks (Raffel et al., 2020; Cheng et al., 2025), typically based on paraphrasing, which avoid this strong assumption but achieve only limited attack success and often distort semantic meaning.

In response to these limitations and to advance the understanding of watermarking vulnerabilities, we propose the *Bias-Inversion Rewriting Attack* (BIRA), motivated by a theoretical analysis of green–red list watermarking showing that reducing the probability of generating green tokens by $\delta > 0$ during the rewriting of watermarked text causes the overall detection probability to decay exponentially in $\delta$. To achieve this, BIRA applies a negative bias to a proxy set of green tokens during paraphrasing with an LLM, without requiring knowledge of the underlying scheme. It consistently evades detection across a wide range of recent watermarking algorithms while preserving the semantics of the original text.

Our contributions are summarized as follows:

- We provide a theoretical characterization of a fundamental vulnerability of current green-red list watermarking schemes.
- Building on this theoretical insight, we introduce BIRA, a query-free attack that weakens the watermark signal by applying a negative logit bias to likely watermarked tokens.
- We conduct extensive experiments demonstrating that BIRA achieves state-of-the-art evasion rates against recent watermarking algorithms while maintaining high semantic fidelity.

## 2 RELATED WORK

**LLM watermarking.** Kirchenbauer et al. (2024a) introduced a widely used green–red list watermarking scheme that partitions the vocabulary into green and red sets and embeds a detectable statistical signal by adding a positive logit bias to green tokens. Subsequent studies have enhanced its robustness by improving key generation and detection (Kirchenbauer et al., 2024b; Liu et al., 2023a; Zhao et al., 2024; Liu et al., 2024; Lee et al., 2024; Lu et al., 2024) or by preserving the original LLM distribution (Wu et al., 2023). Other lines of work investigate sampling-based watermarking approaches (Aaronson & Kirchner, 2022; Hu et al., 2023; Christ et al., 2024).

**Watermark evasion attacks.** Watermark evasion attacks can be broadly categorized under the threat model into two types: *query-based* and *query-free*. Query-based attacks (Jovanović et al., 2024; Chen et al., 2024; Wu & Chandrasekaran, 2024) identify the green token set by issuing a large number of carefully crafted prefix prompts. While effective, this strategy is computationally expensive and has a limited practicality in rate-limited API settings. Additionally, these methods often require knowledge of the watermarking scheme (Wu & Chandrasekaran, 2024) or access to top-$k$ token probabilities of the watermarked model (Chen et al., 2024), which is typically infeasible for proprietary systems.

In contrast, query-free attacks (Kirchenbauer et al., 2024a; Krishna et al., 2023; Cheng et al., 2025; Diaa et al., 2024) avoid these assumptions by operating directly on generated text without interacting with the watermarked model, typically relying on paraphrasing to obscure the statistical signal. Early work (Kirchenbauer et al., 2024a) introduced simple transformation attacks, such as inserting emojis or human-written fragments into watermarked text. More advanced methods either fine-tune an LLM as a paraphrasing expert (Krishna et al., 2023; Diaa et al., 2024) or use a masking-and-rewriting strategy that targets high-entropy tokens and regenerates them with an LLM (Cheng et al., 2025). However, these post-processing approaches often achieve limited attack success, can distort the original meaning, and in some cases require training an additional model. In contrast, our method attains a substantially higher attack success rate while preserving semantic fidelity, without training any extra model in the fully black box scenarios.

## 3 PRELIMINARY

**Language model.** A language model, denoted by $\mathcal{M}$, generates text $y$ by predicting the next token in a sequence. Given an input sequence $x^{0:n-1} = [x^{(0)}, \ldots, x^{(n-1)}]$, the model outputs a logit vector $l^{(n)} = (l_0^{(n)}, \ldots, l_{V-1}^{(n)}) \in \mathbb{R}^V$ from which it derives a probability distribution $Q^{(n)}$ over the vocabulary $\mathcal{V}$ of size $V$ using the softmax operator:

$$Q_u^{(n)} = \frac{\exp(l_u^{(n)})}{\sum_{j=1}^{V} \exp(l_j^{(n)})}, \quad u \in \mathcal{V}.$$

The next token $x^{(n)}$ is then drawn from $Q^{(n)}$, either by sampling or by another decoding strategy.

**Watermarking algorithm.** A watermarking algorithm $\mathcal{W}$ consists of two components: a *generation function* $\mathcal{S}$ and a *detection function* $\mathcal{D}$. Given a secret key $k$, the algorithm $\mathcal{W}_k$ modifies the distribution $Q^{(n)}$ during text generation to produce $\widehat{Q}^{(n)} = \mathcal{M}(x^{0:n-1}, \mathcal{W}_k)$, embedding hidden patterns (e.g., green tokens) into the output $y$. For instance, Kirchenbauer et al. (2024a); Liu et al. (2023a); Zhao et al. (2024); Liu et al. (2024); Lee et al. (2024); Lu et al. (2024) add a positive logit

Figure 1: Illustration of BIRA. A watermarked LLM typically increases the likelihood of sampling green tokens by adding a positive bias $\gamma > 0$ to their logits at each generation step. In contrast, BIRA applies a negative bias $\beta < 0$ to a proxy set of green tokens (since the true set is unknown), thereby suppressing their sampling probability. This inversion lowers the probability of generating green tokens and weakens the watermark signal, enabling the paraphrased text to evade detection.

bias $\gamma > 0$ to $l_u^{(n)}$ for tokens $u \in \mathcal{G}(\mathcal{W}_k)$, the green set generated by the secret key $k$, which increases their sampling probability and biases the generated text $\hat{y}$ toward green tokens. The detection function $\mathcal{D}$ then takes a text sequence $y$ and the same secret key $k$ as input, and determines whether $y$ is watermarked:

$$\mathcal{D}(y, \mathcal{W}_k) = \mathbf{1}\{Z(y; \mathcal{W}_k) \geq \tau\},$$

where $Z(y; \mathcal{W}_k)$ is a test statistic on the watermark patterns (e.g., a one-proportion $z$-statistic on the fraction of green tokens), and $\tau \in \mathbb{R}$ is the detection threshold. Here, the null hypothesis $H_0$ is that the text was not generated with $\mathcal{W}_k$, and the watermark is detected by rejecting $H_0$ when $Z(y; \mathcal{W}_k) \geq \tau$.

**Threat model.** We consider a black-box threat model where the adversary has no knowledge of the watermarking scheme $\mathcal{W}$ or the target model.

**Adversary's objective.** The adversary's goal is to design a text modification function $\mathcal{F}$ that transforms a watermarked text $\hat{y}$ into a modified text $\tilde{y} = \mathcal{F}(\hat{y})$, which is detected as unwatermarked, while preserving the original meaning of $\hat{y}$:

$$\mathcal{F}^* = \arg\min_{\mathcal{F}} \mathbb{E}\left[\mathcal{D}(\tilde{y}, \mathcal{W}_k)\right] \quad \text{s.t.} \quad S(\tilde{y}, \hat{y}) \geq \epsilon, \tag{1}$$

where $S$ is a similarity measure between two texts used to evaluate semantic preservation.

# 4 METHOD

In this section, we first present the theoretical analysis of watermarking vulnerabilities that our attack exploits (Section 4.1), and then describe the attack algorithm (Section 4.2). Figure 1 provides an overview of BIRA.

## 4.1 THEORETICAL ANALYSIS OF WATERMARKING VULNERABILITIES

The goal of a watermark evasion attack is to diminish the overrepresentation of green tokens in a text to a level that cannot be detected statistically. We first show that common watermark detectors, which rely on test statistics like the z-score, are functionally equivalent to a simple threshold test on the empirical green token rate, $\hat{p}(y; \mathcal{W}_k)$ (Theorem 1). Building on this, we prove that if the average probability of generating a green token across the sequence stays below the detection threshold by a margin $\delta$, then the detection probability decreases exponentially in $\delta$ (Theorem 2).

**Theorem 1.** *Let the detector be* $\mathcal{D}(y, \mathcal{W}_k) = \mathbf{1}\{Z(y; \mathcal{W}_k) \geq \tau\}$ *and suppose there exists a nondecreasing function* $h : [0, 1] \to \mathbb{R}$ *with*

$$Z(y; \mathcal{W}_k) = h(\hat{p}(y; \mathcal{W}_k)), \qquad \hat{p}(y; \mathcal{W}_k) = \frac{1}{N} \sum_{n=0}^{N-1} \mathbf{1}\{y^{(n)} \in \mathcal{G}(\mathcal{W}_k)\},$$

where $\mathcal{G}(\mathcal{W}_k)$ denotes the green set produced by watermarking $\mathcal{W}_k$. Then, for a given $N$, there exists $p_\tau \in [0, 1]$ such that

$$\mathcal{D}(y, \mathcal{W}_k) = \mathbf{1}\{\hat{p}(y; \mathcal{W}_k) \geq p_\tau\},$$

with $p_\tau = \inf\{p : h(p) \geq \tau\}$. In particular, for the widely used one-proportion z-test for watermark detection, such a function $h$ exists.

Theorem 1 shows that detection can be expressed in terms of the empirical green rate $\hat{p}(y; \mathcal{W}_k)$ with threshold $p_\tau$. Using this, we now demonstrate that suppressing the average green token probability across the sequence yields exponential decay in the detection probability (Theorem 2).

**Theorem 2.** *Let $\tilde{y} = [\tilde{y}^{(0)}, \ldots, \tilde{y}^{(N-1)}]$ be the attacker's output and let $p_\tau$ be the detection threshold. If there exists $\delta > 0$ such that the average conditional green-token probability satisfies*

$$\frac{1}{N} \sum_{n=1}^{N} \mathbb{E}\Big[ \mathbf{1}\{\tilde{y}^{(n)} \in \mathcal{G}(\mathcal{W}_k)\} \,\Big|\, \tilde{y}^{0:n-1} \Big] \ \leq \ p_\tau - \delta,$$

*then*

$$\Pr\big[\mathcal{D}(\tilde{y}, \mathcal{W}_k) = 1\big] \ \leq \ \exp\left(-\frac{1}{2} N \delta^2\right),$$

Theorem 2 shows that if the average probability of sampling a green token stays at least $\delta$ below the detector threshold $p_\tau$, then the detection probability decays exponentially in $\delta$. In other words, even a small reduction in the green-token probability, when achieved on average over the sequence, is sufficient to make the text statistically undetectable and drive the overall detection probability toward zero. Proofs of Theorems 1 and 2 are provided in Appendix A. Although Theorem 2 is stated for token-level green–red list watermarking, the same analysis extends to recent sentence-level watermarking schemes. The details of the proof and additional experimental results are provided in Appendix E.

**Application to KGW watermarking.** For KGW (Kirchenbauer et al., 2024a), the one-proportion $z$-statistic is $Z(y; W_k) = \big(\hat{p}(y; W_k) - p_0\big)/\sqrt{p_0(1 - p_0)/N}$, where $p_0$ is the predefined green-token ratio and $N$ is the total number of generated tokens. Since $h(p) = \big(p - p_0\big)/\sqrt{p_0(1 - p_0)/N}$ is nondecreasing, the threshold corresponds to $p_\tau = p_0 + \tau\sqrt{p_0(1 - p_0)/N}$. For default setups $p_0 = 0.5$, $\tau = 4$, and $N = 230$, we obtain $p_\tau \approx 0.632$. If the attack keeps $\frac{1}{N}\mathbb{E}\big[\mathbf{1}\{\tilde{y}^{(n)} \in \mathcal{G}(W_k)\} \mid \tilde{y}^{0:n-1}\big] \leq 0.632 - \delta$, then by Theorem 2 the detection probability satisfies $\Pr[D(\tilde{y}, W_k) = 1] \leq \exp(-N\delta^2/2)$; e.g., $\delta = 0.1 \Rightarrow e^{-1.15} \approx 0.316$, $\delta = 0.2 \Rightarrow e^{-4.6} \approx 0.010$.

## 4.2 BIAS-INVERSION REWRITING ATTACK

As established in Theorem 2, successful watermark evasion requires suppressing green token generation across the sequence. In a black box setting, the adversary lacks access to the true green sets $\mathcal{G}(\mathcal{W}_k)$, so we approximate them with a proxy set $\widehat{\mathcal{G}}$. Since watermarking schemes typically embed their signal in low-probability tokens, and token self-information (surprisal) is effective for detecting them Cheng et al. (2025), we identify such tokens using their self-information under a public language model $\mathcal{M}$:

$$I^{(n)} = -\log P_\mathcal{M}(\hat{y}^{(n)} \mid \hat{y}^{0:n-1}).$$

Given a watermarked text $\hat{y} = [\hat{y}^{(0)}, \ldots, \hat{y}^{(N-1)}]$, let $\eta$ be the $q$th percentile of $\{I^{(n)}\}_{n=0}^{N-1}$. The proxy green set is

$$\widehat{\mathcal{G}} \leftarrow \big\{\mathrm{id}(\hat{y}^{(n)}) \,\big|\, I^{(n)} \geq \eta, \ n \in [0, N-1]\big\},$$

where $\mathrm{id}(\cdot)$ maps a token to its vocabulary index. During paraphrase generation $\tilde{y}$, we add a negative logit bias $\beta < 0$ to all tokens in $\widehat{\mathcal{G}}$ at each step $n$:

$$l_u^{(n)} \leftarrow l_u^{(n)} + \beta\, \mathbf{1}\{u \in \widehat{\mathcal{G}}\}.$$

This negative bias suppresses the generation of likely green tokens, which lowers the empirical green rate and thus enables evasion of detection. Since we rely on a proxy green set $\widehat{\mathcal{G}}$, some green tokens may be missed. However, Theorem 2 still holds as long as the average miss rate of the proxy set

---

**Algorithm 1** Pseudocode for Bias-Inversion Rewriting Attack

---

**Require:** System prompt $S$; Watermarked text $\hat{y}^{0:N-1}$; Language model $\mathcal{M}$; Percentile $q \in [0, 1)$; Initial bias $\beta_0 < 0$; lr $> 0$; Max restarts $R$; Max length $L_{\max}$; Window size $h$; threshold $\rho \in (0, 1]$.
                                                    ▷ *Phase 1: Construct Green Token Proxy Set*
1: Compute self-information $I^{(n)}$ for each token $\hat{y}^{(n)}$ using the language model $\mathcal{M}$:
2: **for** $n = 0, \ldots, N-1$ **do**
3:     $I^{(n)} \leftarrow -\log P_{\mathcal{M}}(\hat{y}^{(n)} | \hat{y}^{0:n-1})$
4: **end for**
5: Set percentile threshold $\eta \leftarrow \text{Percentile}(\{I^{(n)}\}_{n=0}^{N-1}, q)$
6: Define the proxy set $\widehat{\mathcal{G}} \leftarrow \{ \text{id}(\hat{y}^{(n)}) \mid I^{(n)} \geq \eta, \ n \in [0, N-1] \}$
                                                   ▷ *Phase 2: Perform Bias-Inversion Rewriting*
7: $\beta \leftarrow \beta_0$
8: **for** $r = 1, \ldots, R$ **do**
9:     Initialize empty sequence $\tilde{y} \leftarrow []$
10:    **for** $t = 0, \ldots, L_{\max} - 1$ **do**
11:       Obtain logits $l^{(t)}$ from $M(\tilde{y}, \hat{y}^{0:N-1}, S)$
12:       Apply negative bias: $l_u^{(t)} \leftarrow l_u^{(t)} + \beta \cdot \mathbf{1}\{u \in \widehat{\mathcal{G}}\}$ for all $u$ in vocabulary
13:       Sample next token $\tilde{y}^{(t)} \sim \text{softmax}(l^{(t)})$
14:       Append $\tilde{y}^{(t)}$ to $\tilde{y}$
15:    **end for**
16:    Let $L$ be the length of $\tilde{y}$.
17:    **if** Distinct-1-Gram-Ratio($\tilde{y}^{L-h:L-1}$) $< \rho$ **then**
18:       $\beta \leftarrow \min(0, \beta + \text{lr})$                ▷ Reduce the strength of bias and restart
19:       Continue
20:    **else**
21:       **return** $\tilde{y}$                                      ▷ Return text
22:    **end if**
23: **end for**
24: **return** $\tilde{y}$

---

is bounded by $\epsilon$, i.e., $\frac{1}{N} \sum_{n=1}^{N} \Pr\left[ \tilde{y}^{(n)} \in \mathcal{G}(\mathcal{W}_k) \setminus \widehat{\mathcal{G}} \mid \tilde{y}^{0:n-1} \right] \leq \varepsilon$ and the average conditional probability of sampling from the proxy set is suppressed: $\frac{1}{N} \sum_{n=1}^{N} \mathbb{E}\left[ \mathbf{1}\{\tilde{y}^{(n)} \in \widehat{\mathcal{G}}\} \mid \tilde{y}^{0:n-1} \right] \leq p'_\tau - \delta$ with $p'_\tau = p_\tau - \varepsilon$. Further details are provided in Appendix A.1.

**Remark.** Our approach differs from the masking-and-rewriting strategy of Cheng et al. (2025), which masks high-entropy tokens and then rewrites the masked spans. In contrast, we apply a negative logit bias to tokens in the proxy green set $\widehat{\mathcal{G}}$ at every decoding step. This consistently reduces the probability of sampling green tokens across the sequence and therefore lowers the detection probability, as established by Theorem 2. Moreover, by avoiding disruptive masking that breaks context, our method better preserves semantic fidelity.

### 4.2.1 MITIGATING TEXT DEGENERATION WITH ADAPTIVE BIAS

We observe that applying a strong negative bias $\beta$ can occasionally cause **text degeneration**, where the model repeatedly generates the same phrase (qualitative examples are provided in Appendix K.2). This arises from a distorted token distribution created by suppressing specific tokens and cannot be resolved by simply regenerating text, since the underlying distribution remains unchanged. To address this, our attack adaptively adjusts $\beta$ by detecting degeneration through monitoring the diversity of the last $h$ generated tokens. Specifically, we compute the distinct 1-gram ratio within this window, $\tilde{y}^{M-h:M-1}$, and classify the text as degenerated if the ratio falls below a predefined threshold $\rho$. The algorithms and details of degeneration detection are provided in Appendix B. Upon detection, the magnitude of negative bias is reduced for the next-generation attempt:

$$\beta \leftarrow \min(0, \beta + \text{lr}),$$

where $\mathrm{lr} > 0$ is a small step size. This adaptive adjustment allows the attack to begin with a strong bias for effective watermark removal and then gracefully reduce its strength only when necessary to prevent semantic degradation. The full procedure of our method is presented in Algorithm 1.

To initialize the logit bias $\beta$, we generate 50 paraphrases from the C4 dataset (Raffel et al., 2020) and gradually decrease $\beta$ (for example, from $-1$ down to $-12$) until degeneration appears in at least one of the 50 outputs. We then use this value as the initial logit bias $\beta_0$, which strengthens the attack while minimizing the risk of degeneration. Since degeneration is rare (2.4% over 500 samples, with an average of only 1.03 iterations per text with $\mathrm{lr} = 0.125$), the computational overhead of the adaptive process is negligible.

## 5 EXPERIMENTS

### 5.1 SETUP

**Dataset.** Following prior work (Kirchenbauer et al., 2024a; Liu et al., 2023a; Zhao et al., 2024; Cheng et al., 2025; Lu et al., 2024), we use the C4 dataset to generate watermarked text. We take the first 500 test samples as prompts and generate 230 tokens for each. Additionally, we evaluate our method on Dolly CW (Conover et al., 2023) and DBpedia (Raffel et al., 2020), with results reported in Appendix D.

**Watermark algorithms.** We evaluate seven recent watermarking methods: KGW (Kirchenbauer et al., 2024a), Unigram (Zhao et al., 2024), UPV (Liu et al., 2023a), EWD (Lu et al., 2024), DIP (Wu et al., 2023), SIR (Liu et al., 2024), and EXP (Aaronson & Kirchner, 2022). For each method, we adopt the default or recommended hyperparameters from the original studies (Pan et al., 2024; Cheng et al., 2025). For KGW, we use a single left hash on only the immediately preceding token to form the green and red token lists, since fewer preceding tokens improve robustness against watermark evasion attacks. Additionally, we evaluate our method against the recent sentence-level watermarking scheme of Dabiriaghdam & Wang (2025). Details of the sentence-level watermarking scheme and the corresponding results are provided in Appendix E.

**Baselines and language models.** We compare against three *query-free* attack baselines: Vanilla (paraphrasing with a language model), DIPPER (Krishna et al., 2023) (a trained paraphrasing expert), and SIRA (Cheng et al., 2025) (masking and rewriting strategy). For DIPPER-1, we set lexical diversity to 60 without order diversity, and for DIPPER-2, we add order diversity of 40 to increase paraphrasing strength.

We evaluate Vanilla, SIRA, and our method on Llama-3.1-8B, Llama-3.1-70B, and GPT-4o-mini with top-p sampling at $0.95$ and temperature $0.7$. The paraphrasing prompt is provided in Appendix I. Since GPT-4o-mini does not expose logits for computing self-information, we use Llama-3.2-3B as an auxiliary model to estimate high self-information tokens. These tokens are first converted into text using the Llama-3.2-3B tokenizer and then re-tokenized with the GPT tokenizer for use in SIRA and our method. Negative logit bias is then applied through the GPT API, which supports token-level logit biasing.

For SIRA, we set the masking threshold to $0.3$, as recommended by Cheng et al. (2025), and apply it to high-entropy tokens across all models. For our method, we use a percentile threshold of $q = 0.5$ to construct the proxy green token set. The initial negative logit bias is $\beta_0 = -4$ for Llama-3.1-8B-Instruct and Llama-3.1-70B-Instruct (Dubey et al., 2023), and $\beta_0 = -11$ for GPT-4o-mini (OpenAI, 2024b), following the initialization strategy in Section 4.2.1, with $\mathrm{lr} = 0.125$.

### 5.1.1 EVALUATION METRICS

We evaluate attacks in terms of both attack efficacy and text quality.

**Attack efficacy.** Our primary measure is the *Attack Success Rate (ASR)*, the proportion of attacked texts for watermarked text misclassified as non-watermarked. Additionally, to mitigate the effect of detector threshold choices, following (Zhao et al., 2024; Liu et al., 2024; Cheng et al., 2025), we build a test set of 500 attacked texts and 500 human-written texts, and adjust the detector's $z$-threshold to match the False Positive Rate (FPR) at 1% and 10%. At these FPRs, we report the corresponding True Positive Rate (TPR) and F1-score.

Table 1: Comparison of watermarking robustness under different attack methods. Our method, BIRA, achieves the highest attack success rate across all baselines.

| Watermark / Attack | KGW | Unigram | UPV | EWD | DIP | SIR | EXP | Avg ASR |
|---|---|---|---|---|---|---|---|---|
| Vanilla (Llama-3.1-8B) | 88.8% | 73.4% | 73.4% | 92.6% | 99.8% | 54.0% | 80.6% | 80.4% |
| Vanilla (Llama-3.1-70B) | 87.4% | 67.0% | 65.0% | 89.4% | 98.8% | 42.8% | 70.4% | 74.4% |
| Vanilla (GPT-4o-mini) | 60.2% | 30.2% | 46.8% | 58.8% | 95.8% | 23.6% | 31.8% | 49.6% |
| DIPPER-1 | 93.8% | 61.2% | 80.6% | 92.8% | 99.4% | 55.6% | 90.8% | 82.0% |
| DIPPER-2 | 97.2% | 71.8% | 85.4% | 96.6% | 99.2% | 70.4% | 97.2% | 88.3% |
| SIRA (Llama-3.1-8B) | 98.8% | 95.0% | 87.6% | 99.8% | 99.6% | 72.8% | 95.2% | 92.7% |
| SIRA (Llama-3.1-70B) | 98.0% | 87.6% | 85.0% | 99.2% | 99.6% | 60.6% | 88.6% | 88.4% |
| SIRA (GPT-4o-mini) | 98.0% | 85.2% | 84.8% | 97.2% | 99.6% | 57.6% | 94.8% | 88.2% |
| **BIRA (Llama-3.1-8B, ours)** | **99.8%** | **99.4%** | **99.8%** | **100.0%** | **100.0%** | **99.6%** | **99.8%** | **99.8%** |
| **BIRA (Llama-3.1-70B, ours)** | **99.4%** | **99.0%** | **99.6%** | **99.8%** | **99.6%** | **98.8%** | **98.0%** | **99.2%** |
| **BIRA (GPT-4o-mini, ours)** | **99.4%** | **100.0%** | **100.0%** | **99.8%** | **99.8%** | **99.4%** | **98.2%** | **99.5%** |

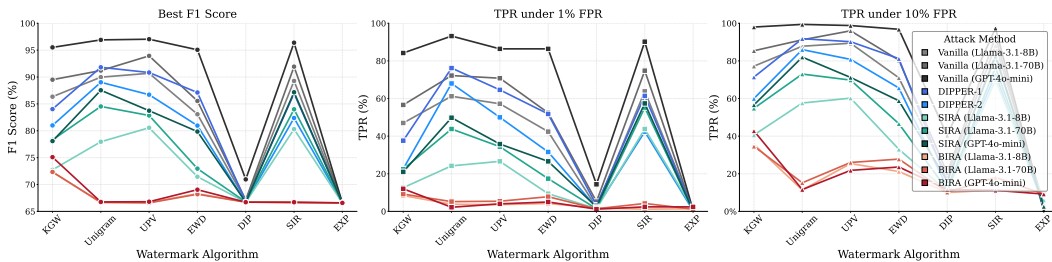

Figure 2: Comparison of detection performance with the adjusted threshold across watermarking algorithms, mitigating the effect of default threshold. We show the best F1 score (↓) and TPR (↓) at FPR of 1% and 10%. BIRA consistently achieves lower F1 and TPR than all baselines, indicating greater difficulty for detectors in distinguishing attacked text from human-written text. Exact values are provided in Appendix L.1.

**Text quality.** We assess text quality using five metrics that cover semantic fidelity, paraphrasing strength, and fluency. To evaluate semantic preservation, we employ three measures. First, we use an **LLM judgement score** (Zheng et al., 2023; Fu et al., 2023; Liu et al., 2023b) from GPT-4o-2024-08-06 (OpenAI, 2024a), which scores meaning preservation on a 1-to-5 scale: a score of 5 indicates perfect fidelity, 4 allows for minor nuances without factual changes, and 3 reflects that only the main idea is preserved while important details or relations are altered (see Appendix H for prompt details). We also compute an **NLI score** using `nli-deberta-v3-large` (He et al., 2020) to assess logical consistency between the original and attacked texts by evaluating mutual entailment. In addition, we report an **S-BERT score** (Reimers & Gurevych, 2019), following (Cheng et al., 2025), which is based on the cosine similarity between sentence embeddings of the two texts.

To quantify the degree of paraphrasing and assess text naturalness, we use two additional metrics. Paraphrasing strength is measured with the **Self-BLEU score** (Zhu et al., 2018), which computes the BLEU score (Papineni et al., 2002) of each attacked text against its corresponding watermarked reference. This measures the overlap between the two texts, where a lower score indicates less lexical overlap and therefore stronger paraphrasing. Text naturalness is evaluated using **Perplexity (PPL)** (Jelinek et al., 1977), where a lower PPL corresponds to more probable and natural text.

## 5.2 EXPERIMENTAL RESULTS

**Attack efficacy.** Table 1 reports the attack success rates of different watermark removal methods across multiple watermarking algorithms. Our method consistently outperforms all baselines across different language models, with especially strong gains against SIR, the most robust existing watermarking algorithm. Notably, on GPT-4o-mini, vanilla paraphrasing attains an average ASR of 49.6, while BIRA reaches 99.5, demonstrating a substantial gain in watermark evasion. To further evaluate effectiveness and reduce the influence of a fixed $z$-threshold, we follow prior work (Zhao et al., 2024; Liu et al., 2024; Cheng et al., 2025) by setting FPR to 1% and 10%, and report the detector's TPR on attacked text using the corresponding adjusted thresholds. We additionally pro-

> **Watermarked Text by KGW ($z$-score: 6.03)**
>
> Graciousness might not seem like the most important thing in defining the success of a nation, but it is paramount for Mr Lim Siong. By the age of 45, he already runs a fortune of close to \$1bn. His wife, Ms Rachel Jia Xu (above), started a chain of popular supermarkets, which now has 13 outlets and employs more than 5,000 people. Not by sheer talent alone, but because of a strong sense of character that has made him a standout amongst the elite among Singapore's business leaders.

> **Attacked Text by BIRA ($z$-score: 0.83)**
>
> Graciousness may appear insignificant in determining what makes a country successful; however, it holds great importance for Mr. Lim Siong. At just 45 years old, he manages nearly \$1 billion in wealth. His spouse, Ms. Rachel Jia Xu, has established an acclaimed supermarket franchise with currently 13 locations employing over 5,000 individuals. His prominence in Singapore's business community is attributed not solely to his exceptional skills but also his remarkable personal qualities.

Figure 3: Qualitative comparison of KGW-watermarked text and the same passage after a BIRA attack with Llama-3.1-8B. The attack paraphrases to suppress green tokens while preserving meaning, lowering the z score from 6.03 to 0.83 and evading detection at a threshold of 4. More examples

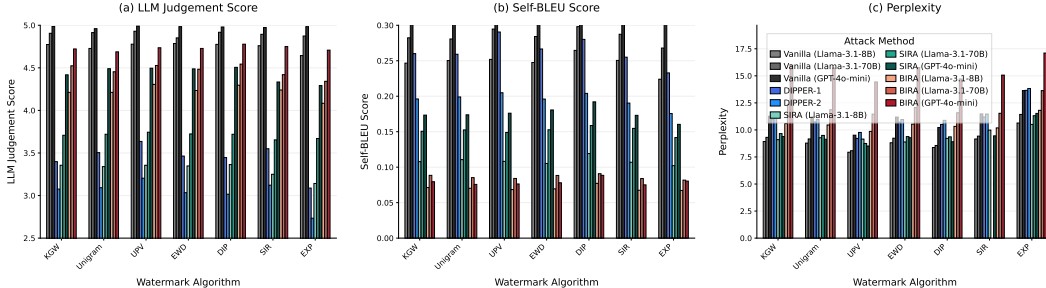

Figure 4: Comparison of text quality across different attacks for various watermarking methods, evaluated by LLM judgment score (↑), Self-BLEU score (↓), and Perplexity (↓). Our method preserves semantic fidelity to the original text compared to other attack baselines (DIPPER and SIRA) while providing stronger paraphrasing, as reflected in lower Self-BLEU scores. Additional results for NLI score (↑) and S-BERT score (↑) are provided in Figure 15 and exact values are detailed in Appendix L.2.

vide the best F1 score each watermarking algorithm can achieve under different attacks. A lower TPR at a given FPR indicates that the detector has greater difficulty distinguishing attacked texts from human-written texts. As shown in Figure 2, our method consistently lies below all baselines, demonstrating its superior attack effectiveness.

**Text quality.** We evaluate the quality of the attacked text using five metrics. As shown in Figure 4, vanilla paraphrasing attains the highest LLM judgment score because its paraphrasing ability is weak and largely preserves the original structure, which leads to low ASR. This is consistent with its highest Self-BLEU score, indicating strong overlap with the source text. In contrast, our method achieves a significantly higher LLM judgment score than stronger baselines such as DIPPER and SIRA, demonstrating better semantic preservation. At the same time, it yields a much lower Self-BLEU score, showing that it generates more diverse paraphrases and relies less on reusing words from the watermarked text. For perplexity, our method remains comparable to other approaches, with only a slight increase when GPT-4o-mini is used. We attribute this to GPT-4o-mini sometimes producing a stiff text that, while grammatically correct and semantically accurate, employs unconventional vocabulary and thus sounds less natural. Qualitative examples illustrating this are provided in Appendix K.3. For NLI and S-BERT scores (Figure 15), the results align with the LLM judgment score and confirm our method's effectiveness.

## 5.3 ABLATION STUDIES AND ANALYSIS

We conduct ablation studies on the logit bias $\beta$ and the percentile $p$ used in our attack, and evaluate the effectiveness of self-information–guided token selection for applying negative logit bias. We also analyze the computational efficiency of different attack methods, and we provide a detection bound

Table 2: Effect of logit bias $\beta$ and percentile $q$ on attack performance

| Logit Bias ($\beta$) | 0.0 | -1.0 | -2.0 | -3.0 | -4.0 | -5.0 | -6.0 | -7.0 | -8.0 | -9.0 |
|---|---|---|---|---|---|---|---|---|---|---|
| ASR ($\uparrow$) | 54.2% | 74.4% | 88.2% | 97.2% | 99.6% | 99.4% | 99.6% | 99.8% | 99.8% | 99.6% |
| LLM Judgment ($\uparrow$) | 4.76 | 4.69 | 4.62 | 4.48 | 4.24 | 3.79 | 3.45 | 3.19 | 3.05 | 2.94 |
| Self-BLEU ($\downarrow$) | 0.25 | 0.20 | 0.15 | 0.11 | 0.07 | 0.04 | 0.03 | 0.02 | 0.01 | 0.01 |
| Perplexity ($\downarrow$) | 9.17 | 8.99 | 9.07 | 9.47 | 10.09 | 11.23 | 12.56 | 13.42 | 14.03 | 14.10 |
| Iteration ($\downarrow$) | 1.00 | 1.00 | 1.00 | 1.01 | 1.03 | 1.20 | 1.48 | 1.83 | 2.54 | 3.36 |
| Percentile ($q$) | 0.0 | 0.1 | 0.2 | 0.3 | 0.4 | 0.5 | 0.6 | 0.7 | 0.8 | 0.9 |
| ASR ($\uparrow$) | 99.0% | 99.6% | 99.2% | 98.6% | 99.6% | 99.6% | 98.4% | 96.4% | 89.0% | 77.2% |
| LLM Judgment ($\uparrow$) | 4.15 | 4.20 | 4.15 | 4.16 | 4.18 | 4.24 | 4.26 | 4.35 | 4.48 | 4.60 |
| Self-BLEU ($\downarrow$) | 0.06 | 0.06 | 0.06 | 0.06 | 0.06 | 0.07 | 0.08 | 0.10 | 0.13 | 0.18 |
| Perplexity ($\downarrow$) | 12.16 | 12.01 | 11.72 | 11.37 | 10.68 | 10.09 | 9.62 | 9.30 | 8.95 | 8.93 |
| Iteration ($\downarrow$) | 1.06 | 1.06 | 1.05 | 1.06 | 1.05 | 1.03 | 1.02 | 1.02 | 1.01 | 1.00 |

analysis that validates our Theorem 2 in the Appendix C. Unless otherwise specified, all experiments are performed on the Llama-3.1-8B-Instruct model with the SIR watermarking method, following the setup in Section 5.1.

**Effect of logit bias $\beta$ and percentile $q$.** We vary $\beta$ from 0.0 to $-9.0$ with the percentile fixed at $q = 0.5$. Table 2 shows that without logit bias ($\beta = 0.0$, equivalent to vanilla paraphrasing), the ASR is low, but it increases as the absolute value of $\beta$ grows. This is consistent with Theorem 2: increasing the negative bias further suppresses green token sampling and thus lowers the overall probability of detection. However, larger negative values of $\beta$ gradually degrade text quality and require more iterations, as excessive bias restricts the token distribution too strongly.

Next, we vary $q$ from 0.0 to 0.9 with $\beta = -4.0$. Table 2 shows that when $q = 0$ (bias applied to all tokens in the watermarked text), ASR is moderately high, but text quality degrades slightly, and the number of iterations increases because many tokens are suppressed. As $q$ grows, the proxy set contains fewer tokens and fewer are suppressed, so ASR drops since the watermark signal is not effectively removed, while text quality improves as the token distribution is less constrained.

**Effectiveness of statistical signal suppression.** Table 3 presents $z$-scores and corresponding detection thresholds $\tau$ for different attacks under the SIR and Unigram watermarking schemes. Our method achieves lower $z$-scores than all baselines, making detection substantially more difficult. A lower $z$-score indicates that the attacked text is harder to distinguish from human-written text. Additional results for other watermarking schemes are presented in Table 9, where our method consistently outperforms all baselines.

**Impact of token selection.** To evaluate the effectiveness of self-information–guided token selection when applying logit bias, we vary the selection ratio from 0.1 to 0.9, choosing the highest self-information tokens at each ratio and comparing against a random selection. As shown in Figure 5, applying negative logit bias to self-information-guided tokens consistently outperforms random selection, demonstrating its effectiveness in constructing a proxy green set $\widehat{\mathcal{G}}$.

**Computational efficiency.** To assess computational overhead, we measured the average execution time per attack over 500 samples using the KGW watermark under the setup in Section 5.1. All experiments were conducted on a single A6000 GPU, except for DIPPER built on T5-XXL (Raffel et al., 2020), which required two GPUs. As shown in Table 4, Vanilla is the most efficient baseline since it introduces no additional overhead. BIRA is the next most efficient, though it exhibits higher variance. This is caused by its adaptive bias procedure, which is designed to prevent text degeneration.

Table 3: $z$-score comparison of attacks on SIR and Unigram watermarking scheme.

| Watermark | SIR ($\tau = 0.2$) | Unigram ($\tau = 4.0$) |
|---|---|---|
| Vanilla | $0.19 \pm 0.10$ | $3.04 \pm 1.52$ |
| DIPPER-1 | $0.18 \pm 0.11$ | $3.63 \pm 1.65$ |
| DIPPER-2 | $0.14 \pm 0.11$ | $3.10 \pm 1.63$ |
| SIRA | $0.14 \pm 0.12$ | $1.63 \pm 1.38$ |
| BIRA | $-0.06 \pm 0.09$ | $-0.34 \pm 1.61$ |

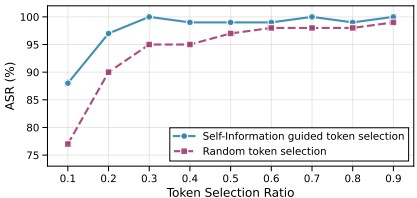

Figure 5: Comparison of ASR for self-information–guided token selection and random token selection.

Table 4: Average execution time (in seconds) for different attacks.

| Attack | Time |
|---|---|
| Vanilla | $5.52 \pm 2.82$ |
| DIPPER | $9.73 \pm 2.24$ |
| SIRA | $8.57 \pm 2.03$ |
| BIRA | $7.95 \pm 9.01$ |

This procedure was triggered in only 2.6% of samples, and those rare cases had a much longer average runtime of 66.81 seconds because repeated generation continued until the maximum length is reached. By contrast, the vast majority of samples (97.4%) completed in a single iteration with an average of 6.38 seconds, accounting for BIRA's overall efficiency despite the variance introduced by a few outliers.

## 6 CONCLUSION

This paper exposes fundamental vulnerabilities in LLM watermarking through a theoretical analysis, from which we developed the Bias-Inversion Rewriting Attack (BIRA). Our attack erases the watermark's statistical signal by applying a negative logit bias to tokens identified using self-information. We empirically demonstrate that BIRA consistently evades detection from recent watermarking schemes while preserving the original text's meaning. Our work reveals significant limitations in current methods, highlighting the need for more rigorous evaluation of watermarking and motivating the defenses that remain robust against sophisticated paraphrasing attacks.

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

## A    PROOF OF THEOREM

*Proof of Theorem 1.* Fix $N \in \mathbb{N}$. By assumption there exists a nondecreasing $h : [0,1] \to \mathbb{R}$ with

$$Z(y; \mathcal{W}_k) = h(\hat{p}(y; \mathcal{W}_k)), \qquad \hat{p}(y; \mathcal{W}_k) = \frac{1}{N} \sum_{n=0}^{N-1} \mathbf{1}\{y_n \in \mathcal{G}(\mathcal{W}_k)\}.$$

The range of $\hat{p}$ is the grid $\mathcal{P}_N := \{0, 1/N, \dots, 1\}$. Define

$$p_\tau = \min\{ p \in \mathcal{P}_N : h(p) \geq \tau \},$$

taking $p_\tau = 1$ if the set is empty and $p_\tau = 0$ if $h(p) \geq \tau$ for all $p \in \mathcal{P}_N$. Since $h$ is nondecreasing, for any $p, p' \in \mathcal{P}_N$ with $p \geq p_\tau > p'$ we have $h(p) \geq h(p_\tau) \geq \tau$ while $h(p') < \tau$. Therefore, for any $y$,

$$\mathcal{D}(y, \mathcal{W}_k) = \mathbf{1}\{Z(y; \mathcal{W}_k) \geq \tau\} = \mathbf{1}\{h(\hat{p}(y; \mathcal{W}_k)) \geq \tau\} = \mathbf{1}\{\hat{p}(y; \mathcal{W}_k) \geq p_\tau\}.$$

$\square$

*Proof of Theorem 2.* By Theorem 1, for fixed $N$ there exists $p_\tau$ with $\mathcal{D}(y, \mathcal{W}_k) = \mathbf{1}\{\hat{p}(y; \mathcal{W}_k) \geq p_\tau\}$. Define the indicator variables and their conditional expectations:

$$X_n := \mathbf{1}\{\tilde{y}^{(n)} \in \mathcal{G}(\mathcal{W}_k)\}, \qquad p_n := \mathbb{E}[X_n \mid \mathcal{F}_{n-1}],$$

where $\mathcal{F}_n := \sigma(\tilde{y}^{0:n-1})$ is the natural filtration. The premise of the theorem is that the average conditional probability $\bar{p}_N := \frac{1}{N} \sum_{n=1}^{N} p_n$ satisfies $\bar{p}_N \leq p_\tau - \delta$.

Define the martingale difference sequence

$$D_n := X_n - p_n,$$

and the martingale

$$M_N := \sum_{n=1}^{N} D_n.$$

This is a martingale difference sequence since $\mathbb{E}[D_n \mid \mathcal{F}_{n-1}] = \mathbb{E}[X_n \mid \mathcal{F}_{n-1}] - p_n = p_n - p_n = 0$. Moreover, since $X_n \in \{0,1\}$ and $p_n \in [0,1]$, the increments are bounded in the interval $D_n \in [-1,1]$.

We can relate the empirical green rate $\hat{p}(\tilde{y}; \mathcal{W}_k)$ to the martingale $M_N$:

$$\hat{p}(\tilde{y}; \mathcal{W}_k) = \frac{1}{N} \sum_{n=1}^{N} X_n = \frac{1}{N} \sum_{n=1}^{N} (D_n + p_n) = \frac{M_N}{N} + \bar{p}_N.$$

Thus, the detection event occurs iff:

$$\hat{p}(\tilde{y}; \mathcal{W}_k) \geq p_\tau \iff \frac{M_N}{N} + \bar{p}_N \geq p_\tau \iff M_N \geq N(p_\tau - \bar{p}_N).$$

Using the premise that $\bar{p}_N \leq p_\tau - \delta$, we have $p_\tau - \bar{p}_N \geq \delta$. Therefore,

$$\Pr(\hat{p}(\tilde{y}; \mathcal{W}_k) \geq p_\tau) \leq \Pr(M_N \geq N\delta).$$

By the Azuma–Hoeffding inequality, for increments $D_n$ bounded in an interval of range $1 - (-1) = 2$,

$$\Pr(M_N \geq N\delta) \leq \exp\left(-\frac{2(N\delta)^2}{\sum_{n=1}^{N} 2^2}\right) = \exp\left(-\frac{2N^2\delta^2}{4N}\right) = \exp\left(-\frac{N\delta^2}{2}\right),$$

which yields the claim. $\square$

## A.1 ROBUSTNESS TO PROXY GREEN SETS

In the black-box setting, the adversary does not have access to the true green sets $\mathcal{G}(\mathcal{W}_k)$, so we use a proxy $\widehat{\mathcal{G}}$ that may not contain all green tokens. The guarantee of Theorem 2 still holds if the following two conditions are met.

*Average miss rate bound.* The average probability that a sampled token is a true green token not included in the proxy is at most $\varepsilon$:

$$\frac{1}{N} \sum_{n=1}^{N} \Pr\left(\tilde{y}^{(n)} \in \mathcal{G}(\mathcal{W}_k) \setminus \widehat{\mathcal{G}} \,\Big|\, \tilde{y}^{0:n-1}\right) \leq \varepsilon.$$

*Average proxy suppression.* The attack suppresses tokens from the proxy on average, such that

$$\frac{1}{N} \sum_{n=1}^{N} \mathbb{E}\left[\mathbf{1}\{\tilde{y}^{(n)} \in \widehat{\mathcal{G}}\} \,\Big|\, \tilde{y}^{0:n-1}\right] \leq p'_\tau - \delta \quad \text{for some } \delta > 0,$$

where $p'_\tau := p_\tau - \varepsilon$.

Under these conditions, the conclusion of Theorem 2 still holds. This follows by showing that its premise is satisfied. Let $\bar{p}_N$ be the average conditional green probability:

$$\begin{aligned}
\bar{p}_N &= \frac{1}{N} \sum_{n=1}^{N} \mathbb{E}\left[\mathbf{1}\{\tilde{y}^{(n)} \in \mathcal{G}(\mathcal{W}_k)\} \,\Big|\, \tilde{y}^{0:n-1}\right] \\
&= \frac{1}{N} \sum_{n=1}^{N} \mathbb{E}\left[\mathbf{1}\{\tilde{y}^{(n)} \in \widehat{\mathcal{G}}\} \,\Big|\, \tilde{y}^{0:n-1}\right] + \frac{1}{N} \sum_{n=1}^{N} \mathbb{E}\left[\mathbf{1}\{\tilde{y}^{(n)} \in \mathcal{G}(\mathcal{W}_k) \setminus \widehat{\mathcal{G}}\} \,\Big|\, \tilde{y}^{0:n-1}\right] \\
&\leq (p'_\tau - \delta) + \varepsilon \\
&= (p_\tau - \varepsilon - \delta) + \varepsilon = p_\tau - \delta.
\end{aligned}$$

This satisfies the premise of Theorem 2, so the result holds.

## B DETAILS OF THE TEXT DEGENERATION DETECTION FUNCTION

---
**Algorithm 2** Text Degeneration Detection

---
**Require:** Paraphrased text $\tilde{y} = [\tilde{y}^{(0)}, \ldots, \tilde{y}^{(L-1)}]$; collapse window $h \in \mathbb{N}$; collapse threshold $\rho \in (0,1]$.
 1: $m \leftarrow \min(h, L)$
 2: $W \leftarrow [\tilde{y}^{(L-m)}, \ldots, \tilde{y}^{(L-1)}]$                                   ▷ last $m$ tokens
 3: $U \leftarrow \{\text{id}(u) \mid u \in W\}$                               ▷ set of distinct token ids
 4: **if** $|W| = 0$ **then**
 5:     **return** False
 6: **end if**
 7: $r \leftarrow |U|/|W|$                                ▷ distinct one gram ratio in the window
 8: **if** $r < \rho$ **then**
 9:     **return** True                              ▷ degeneration detected
10: **else**
11:     **return** False
12: **end if**

---

For paraphrasing, we set the maximum generation length of the LLM to 1,500 tokens. We observed that paraphrased text is typically generated normally when no degeneration occurs. However, when degeneration does occur, the text begins normally but then suddenly repeats the same phrase until the maximum token limit is reached, as shown in Appendix K.2. In the degenerated samples we examined, both the starting point of the repetition and the length of the repeated phrase varied. To ensure a sufficient detection window, we chose a large maximum generation length and a window

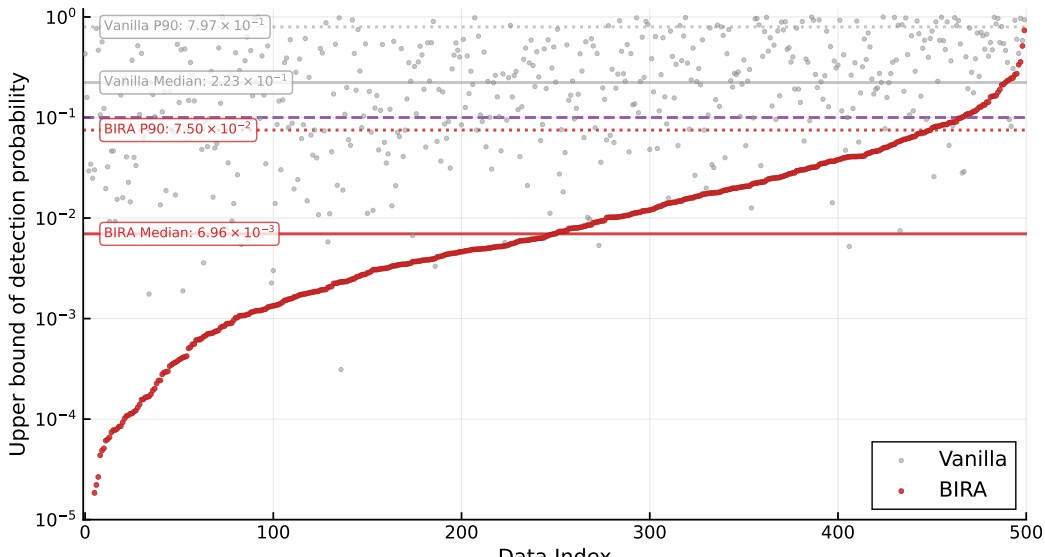

Figure 6: Detection upper bounds per sample from Theorem 2, sorted by BIRA. BIRA substantially reduces the upper bound on detection probability for most samples compared to Vanilla.

size of $h = 450$ tokens. We set the threshold $\rho = 0.25$, meaning that if more than 75% of the tokens within the detection window are duplicates (which is not normal for natural text), the text is considered largely repetitive and redundant.

## C  DETECTION BOUND ANALYSIS

To validate Theorem 2, we compute the per-sample upper bound on the detection probability under the Unigram watermark for 500 samples, using the experimental setup described in Section 5.1. For ease of analysis, we generate watermarked text with Llama-3.2-3B and apply the BIRA attack with Llama-3.1-8B, since the two models share the same tokenizer. For each attacked sequence of length $N$, we calculate the average conditional green probability

$$\bar{p} = \frac{1}{N} \sum_{n=1}^{N} \mathbb{E}\Big[ \mathbf{1}\{\tilde{y}^{(n)} \in \mathcal{G}(\mathcal{W}_k)\} \,\Big|\, \tilde{y}^{0:n-1} \Big],$$

set $\hat{\delta} = \max\{0,\, p_\tau - \bar{p}\}$, and evaluate the bound $\exp\big(-\frac{1}{2}N\hat{\delta}^2\big)$. Figure 6 shows that BIRA yields much lower per-sample detection bounds than Vanilla for most samples. At the 90th percentile, the upper bound for BIRA is $-7.50 \times 10^{-2}$, whereas for Vanilla it is $-2.23 \times 10^{-1}$.

Table 5: Comparison of watermarking robustness under different attack methods on the Dolly CW dataset.

| Attack \ Watermark | KGW | Unigram | UPV | EWD | DIP | SIR | EXP | Avg ASR |
|---|---|---|---|---|---|---|---|---|
| Vanilla (Llama-3.1-8B) | 81.0% | 65.0% | 70.0% | 78.0% | 99.0% | 49.0% | 68.0% | 72.9% |
| DIPPER-1 | 94.0% | 61.0% | 73.0% | 96.0% | 99.0% | 62.0% | 87.0% | 81.7% |
| DIPPER-2 | 98.0% | 74.0% | 83.0% | 94.0% | 100.0% | 66.0% | 95.0% | 87.1% |
| SIRA (Llama-3.1-8B) | 97.0% | 91.0% | 90.0% | 99.0% | 99.0% | 73.0% | 94.0% | 91.9% |
| **BIRA (Llama-3.1-8B, ours)** | **100.0%** | **99.0%** | **99.0%** | **100.0%** | **99.0%** | **98.0%** | **97.0%** | **98.9%** |

Table 6: Comparison of watermarking robustness under different attack methods on the DBPEDIA Class dataset.

| Attack \ Watermark | KGW | Unigram | UPV | EWD | DIP | SIR | EXP | Avg ASR |
|---|---|---|---|---|---|---|---|---|
| Vanilla (Llama-3.1-8B) | 76.0% | 63.0% | 61.0% | 83.0% | 97.0% | 46.0% | 64.0% | 70.0% |
| DIPPER-1 | 87.0% | 52.0% | 75.0% | 92.0% | 100.0% | 67.0% | 83.0% | 79.4% |
| DIPPER-2 | 93.0% | 63.0% | 76.0% | 95.0% | 100.0% | 80.0% | 96.0% | 86.1% |
| SIRA (Llama-3.1-8B) | 95.0% | 90.0% | 90.0% | 99.0% | 97.0% | 71.0% | 88.0% | 90.0% |
| **BIRA (Llama-3.1-8B, ours)** | **98.0%** | **100.0%** | **96.0%** | **99.0%** | **100.0%** | **99.0%** | **95.0%** | **98.1%** |

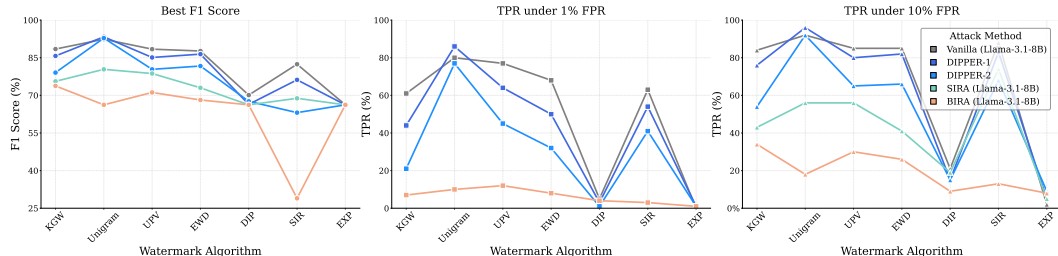

Figure 7: Comparison of detection performance with adjusted thresholds across watermarking algorithms on the Dolly CW dataset. We show the best F1 score (↓) and TPR (↓) at FPR of 1% and 10%.

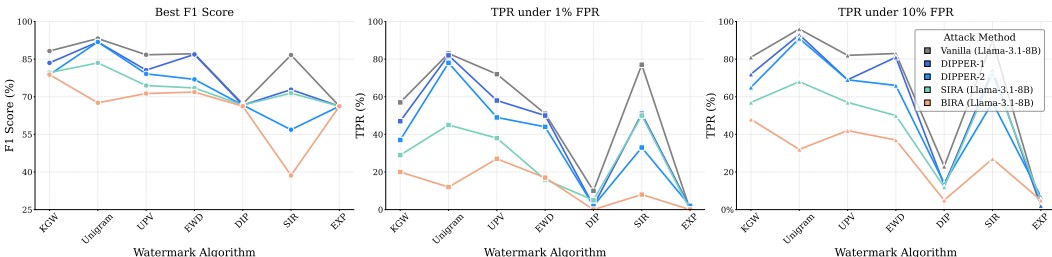

Figure 8: Comparison of detection performance with adjusted thresholds across watermarking algorithms on the DBPEDIA dataset. We show the best F1 score (↓) and TPR (↓) at FPR of 1% and 10%.

## D    EXTENDED EXPERIMENTAL RESULTS ON ADDITIONAL DATASETS

To strengthen the evaluation of our method, we conduct additional experiments on two datasets, Dolly CW Conover et al. (2023) and DBPEDIA Class Raffel et al. (2020), using 100 examples and the experimental setup described in Section 5.1 with the Llama 3.1 8B model. We evaluate attack performance using the default hyperparameters, report TPR at FPR levels of 1% and 10% to mitigate the effect of a fixed threshold, and include text quality evaluation.

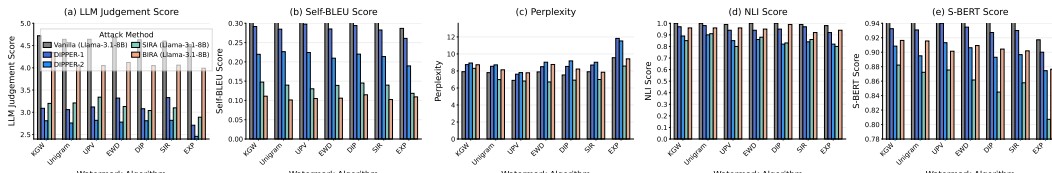

Figure 9: Comparison of text quality across different attacks for various watermarking methods, evaluated using LLM judgment score ($\uparrow$), Self-BLEU ($\downarrow$), and Perplexity ($\downarrow$) on the Dolly CW dataset.

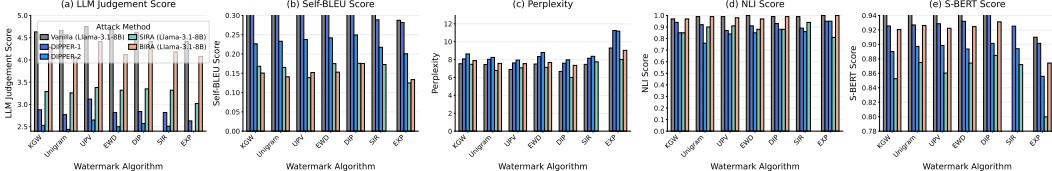

Figure 10: Comparison of text quality across different attacks for various watermarking methods, evaluated using LLM judgment score ($\uparrow$), Self-BLEU ($\downarrow$), and Perplexity ($\downarrow$) on the DBPEDIA dataset.

Figure 7 and Figure 8 show that our method consistently outperforms the baselines under the default settings. Figure 9 and Figure 10 further demonstrate that our method achieves superior performance under the adaptive threshold setup.

For text quality evaluation, our method preserves semantic fidelity more effectively than the strong baselines DIPPER and SIRA across the five metrics, consistent with the trends observed in the main results (Figure 4).

## E APPLICABILITY TO SENTENCE-LEVEL WATERMARKING

Our method is primarily designed for token-level watermarking schemes, since they are fundamental and dominant paradigms. To strengthen robustness against paraphrasing, recent works Hou et al. (2024); Dabiriaghdam & Wang (2025) have proposed more robust sentence-level watermarking schemes, albeit with much higher computational cost. However, these schemes are conceptually similar to token-level watermarking, so the core principle behind BIRA naturally extends to these sentence-level methods as well.

Conceptually, sentence-level watermarking schemes Hou et al. (2024); Dabiriaghdam & Wang (2025) replace the "green/red" token distinction with a "valid/invalid" region in sentence embedding space. Let $s_n$ denote the $n$-th sentence and $e_n = f_{\text{em}}(s_n)$ its embedding under an embedding model $f_{\text{em}}$. During generation, these methods repeatedly regenerate $s_{n+1}$ until the similarity

$$\text{sim}(e_n, e_{n+1})$$

falls within a predefined interval $[a, b]$, and detection is based on counting how many sentence pairs satisfy this validity condition. This is directly analogous to token-level watermarks that count how many tokens fall into a green token set.

**Extension of Theorem 2 to Sentence-Level Watermarking.** Our Theorem 2 can be extended by redefining the random variable from an indicator of sampling a green token to an indicator of a valid sentence pair. Specifically, instead of

$$Z_n = \mathbf{1}\{\tilde{y}^{(n)} \in \mathcal{G}(\mathcal{W}_k)\},$$

we consider

$$Z_n = \mathbf{1}\big[\text{sim}(f_{\text{em}}(s_n), f_{\text{em}}(s_{n+1})) \in [a, b]\big],$$

and the detector thresholds the empirical average

$$\hat{p}_{\text{sent}} = \frac{1}{N} \sum_{n=1}^{N} Z_n.$$

Table 7: Comparison of ASR and text quality across different attacks against SimMark sentence-level watermarking.

| Method | ASR ($\uparrow$) | LLM Judgement ($\uparrow$) | Self-BLEU ($\downarrow$) | PPL ($\downarrow$) | NLI ($\uparrow$) | S-BERT ($\uparrow$) |
|---|---|---|---|---|---|---|
| Vanilla (Llama-3.1-8B) | 0.544 | 4.752 | 0.228 | 8.191 | 0.958 | 0.904 |
| DIPPER-1 | 0.054 | 3.454 | 0.332 | 8.876 | 0.930 | 0.908 |
| DIPPER-2 | 0.082 | 3.036 | 0.238 | 9.505 | 0.856 | 0.878 |
| SIRA (Llama-3.1-8B) | 0.872 | 3.450 | 0.102 | 7.988 | 0.728 | 0.830 |
| BIRA (Llama-3.1-8B, ours) | 0.826 | 4.110 | 0.060 | 9.665 | 0.862 | 0.857 |

For the rewriting attack, let $\tilde{s}_n$ denote the rewritten sentences and let $\tilde{f}_{\text{em}}$ denote an auxiliary embedding. We then define

$$\tilde{Z}_n = \mathbf{1}\big[\text{sim}(\tilde{f}_{\text{em}}(\tilde{s}_n), \tilde{f}_{\text{em}}(\tilde{s}_{n+1})) \in [\tilde{a}, \tilde{b}]\big].$$

During rewriting, we repeatedly generate paraphrases $\tilde{s}_{n+1}$ until they satisfy

$$\text{sim}\big(\tilde{f}_{\text{em}}(\tilde{s}_n), \tilde{f}_{\text{em}}(\tilde{s}_{n+1})\big) \notin [\tilde{a}, \tilde{b}],$$

which effectively reduces the probability that $\tilde{Z}_n = 1$. Following the same logic as in Theorem 2, ensuring

$$\frac{1}{N} \sum_{n=1}^{N} \mathbb{E}[\tilde{Z}_n] \leq p_\tau - \delta$$

for some $\delta > 0$ yields an analogous exponential upper bound on the detector's success probability.

Thus, although our current theorem and implementation operate at the token level, the theoretical framework and BIRA's principle extend naturally to sentence-level watermarking. An interesting direction for future work is to adapt our method to sentence-level watermarking in practice, where a key question is how well the adversary's embedding model $\tilde{f}_{\text{em}}$ aligns with the embedding model $f_{\text{em}}$ used in generating watermarked text, and how to estimate the unknown validity interval $[\tilde{a}, \tilde{b}]$.

**Experimental results of BIRA on sentence-level watermarking.** To evaluate the effectiveness of our token-level BIRA attack against sentence-level watermarking, we conduct experiments under the setup in Section 5.1 using the Llama-3.1-8B model with the SimMark watermarking scheme Dabiriaghdam & Wang (2025), which is a state-of-the-art sentence-level watermark. For detection, we use a $z$-score threshold of $5.03$, which yields an FPR of $1\%$, and adopt the hyperparameter setting recommended by Dabiriaghdam & Wang (2025), using the interval $[0.68, 0.76]$ for cosine similarity on the C4 dataset.

Table 7 shows that BIRA still achieves strong attack performance, albeit slightly weaker than on token-level watermarking schemes. We attribute this to BIRA's strengthened paraphrasing ability, which alters token-level choices sufficiently to shift sentence embeddings outside the valid region. Interestingly, DIPPER-1 and DIPPER-2 exhibit very weak attack performance under SimMark, highlighting the need for stronger stress-testing methods for evaluation, while SIRA attains the highest ASR but degrades text quality more than BIRA. We hypothesize that SIRA's performance gain partly arises from generating lower-quality paraphrases whose sentence embeddings deviate further from the original, whereas BIRA achieves a more favorable trade-off between watermark evasion and semantic fidelity.

> **Human written text in C4 dataset with PPL 25.4**
>
> Yes, we all have separate inboxes. There is no giant inbox where all messages to moderators go. So if we either don't understand a PM we get from a mod or admin we're allowed to ask in PM? Am I allowed to say I have a really hard time reading really long PMs? What I describe as motorway length PMs cause I get halfway then my brain turns it to word salad and goes into information overload? Maybe you need to clarify the rule so not wrong things can be read into it. I wish I had been that lucky. I always get anxiety when I get pm from a monitor. Ok! I didn't know that! Exactly I just discover this myself! Don't let Turtleboy fool you! He's bad to the bone! I know many on the boards here are very sensitive & struggle with anxiety, but everyone has a completely anonymous screen name. No one knows who "emgreen"

Figure 11: An example of human written text in the C4 dataset with high PPL

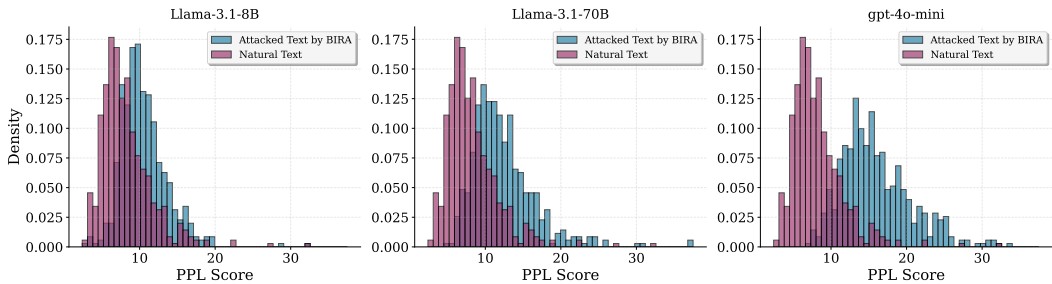

Figure 12: Overlap between the PPL distributions of natural text and BIRA-attacked text for KGW watermarking.

# F  ANALYSIS OF PERPLEXITY INCREASE UNDER BIRA

In this section, we analyze two concerns raised by the increased PPL under the BIRA attack. While our method slightly increases the PPL of attacked text for Llama-3.1-8B and Llama-3.1-70B, the PPL for GPT-4o-mini increases more noticeably. This raises two questions: (1) does this increase lead to a perceptible degradation of readability in real-world scenarios, and (2) can PPL be used as a post-hoc signal to flag attacked text?

**Human evaluation of attacked text with high PPL.** Qualitative analysis in Appendix K.3 shows that attacked texts with high PPL remain grammatical and reasonably readable despite their elevated perplexity scores. Our evaluation suggests that this PPL increase may stem from stylistic factors such as overuse of hyphens or particular synonym choices. Crucially, we also observe that human-written text from the C4 dataset can exhibit similarly high PPL, particularly in informal or conversational sentences (Figure 11). This supports the view that PPL is not a perfect proxy for human-perceived quality in real-world scenarios.

**Can PPL be used as a post-hoc signal to flag attacked text?** Perplexity alone is not a sufficiently reliable detection signal, due to its significant overlap between natural and attacked text. As illustrated in Figure 12, the PPL distributions of attacked text by BIRA for KGW watermarking and natural text in C4 dataset overlap substantially for Llama-3.1-8B and Llama-3.1-70B, making PPL ineffective for discriminating between them. For GPT-4o-mini, the BIRA distribution is more shifted toward higher PPL, but there is still considerable overlap with natural text. We suspect that part of this shift is caused by the mismatch between GPT-4o-mini's tokenizer and the auxiliary tokenizer (Llama-3.1-3B) used to evaluate token self-information in our query-free attack setup.

Additionally, unlike statistical watermarking, which provides formal guarantees via a calibrated test statistic, PPL-based detection has no such theoretical grounding. Thus, although designing BIRA variants that lower PPL while preserving attack strength is interesting future work, perplexity remains a heuristic and lacks the robustness needed for a reliable post-hoc detection mechanism.

Table 8: Comparison of watermarking robustness under different attack methods with ADA fine-tuned on KGW watermark.

| Watermark / Attack | KGW | Unigram | UPV | EWD | DIP | SIR | EXP | Avg ASR |
|---|---|---|---|---|---|---|---|---|
| Vanilla (Llama-3.1-8B) | 88.8% | 73.4% | 73.4% | 92.6% | 99.8% | 54.0% | 80.6% | 80.4% |
| Dipper-1 | 93.8% | 61.2% | 80.6% | 92.8% | 99.4% | 55.6% | 90.8% | 82.0% |
| Dipper-2 | 97.2% | 71.8% | 85.4% | 96.6% | 99.2% | 70.4% | 97.2% | 88.3% |
| ADA (Llama-3.1-8B) | 98.0% | 90.2% | 88.4% | 99.2% | 100.0% | 71.0% | 96.6% | 91.9% |
| SIRA (Llama-3.1-8B) | 98.8% | 95.0% | 87.6% | 99.8% | 99.6% | 72.8% | 95.2% | 92.7% |
| **BIRA (Llama-3.1-8B, ours)** | **99.8%** | **99.4%** | **99.8%** | **100.0%** | **100.0%** | **99.6%** | **99.8%** | **99.8%** |

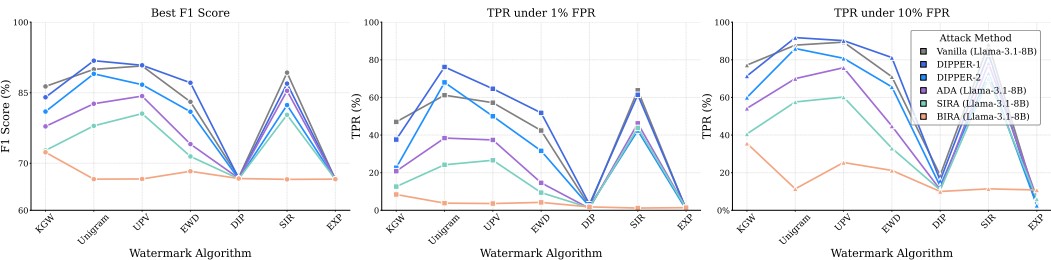

Figure 13: Comparison of detection performance across watermarking algorithms with ADA fine-tuned on the KGW watermark. We show the best F1 score (↓) and TPR (↓) at FPR of 1% and 10%.

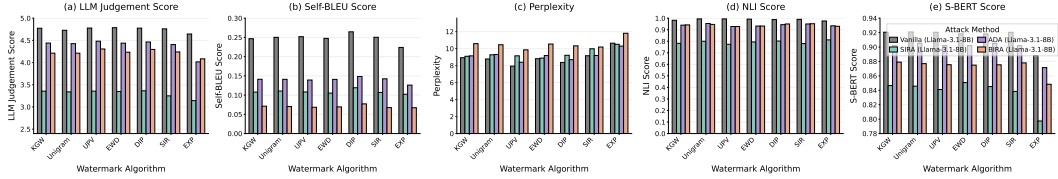

Figure 14: Comparison of text quality across different attacks with ADA fine-tuned on the KGW watermark, evaluated using LLM judgment score (↑), Self-BLEU (↓), and Perplexity (↓).

# G  COMPARISON WITH ADAPTIVE ATTACKS

In this work, our main evaluation focuses on query-free attacks without knowledge of the watermarking scheme. To further assess the effectiveness of our method, we also compare against the adaptive attack (ADA) of Diaa et al. (2024), which operates in a less restrictive threat model: ADA assumes the watermarking scheme is known, fine-tunes a paraphrasing model specifically to that watermark, and then relies on transferability to unseen watermarking schemes. For this comparison, we follow the setup in Section 5.1 using Llama-3.1-8B fine-tuned on KGW watermarking for ADA.

Table 8 and Figure 13 show that SIRA and BIRA outperform ADA in attack success, even though they operate under a more restrictive threat model. For text quality, however, Figure 14 shows that ADA attains higher semantic fidelity in terms of LLM judgment score. These results indicate that BIRA is a substantially stronger attack, with a slight trade-off in semantic fidelity.

## H PROMPT FOR SEMANTIC JUDGMENT (GPT)

---

**LLM-as-a-Judge Prompt**

You are an impartial evaluator.

You will receive: 1) **ORIGINAL**: the source text 2) **PARAPHRASE**: a rewritten version of the original

**Your task:** Judge how well the PARAPHRASE preserves the ORIGINAL's semantic meaning. Ignore style, tone, formality, phrasing, length, and order of information.

**What to check (do not output your analysis):** - Core propositions and claims are preserved. - Entities, numbers, dates, units, polarity/negation, modality, and causal/temporal relations match. - No contradictions; no key facts dropped or altered. - Added details that do not change meaning should not be penalized.

**Rating scale (1–5):** [5] Complete preservation — bidirectional entailment; no contradictions; all key facts preserved. [4] High preservation — meaning preserved with at most minor nuances/omissions; no factual changes. [3] Moderate — main idea preserved, but some important details/relations are missing or changed. [2] Low — only basic topic overlaps; several important changes/omissions or contradictions. [1] Minimal/none — meaning is fundamentally different or unrelated.

**Special cases:** - If specificity differs, score based on whether the essential meaning is preserved. - Any mismatch in numbers, named entities, dates, or negation is a serious error.

**Output Format:** - Provide *only* a number in square brackets. No explanations, additional text, or commentary. - Correct format: [3] - Incorrect format: "The score is [3]" | "I rate this [3]" | "[3] - good preservation"

---

**User Prompt:**

---

Here's the ORIGINAL: [Watermarked text]

Here's the PARAPHRASE: [Attack text]

---

## I  PARAPHRASING PROMPT

---

**Paraphrasing Prompt**

You are an expert paraphrasing system designed to rewrite text while maintaining complete semantic fidelity. Follow these precise guidelines:

**CORE REQUIREMENTS:** - Preserve 100% of the original meaning and information content - Maintain identical factual accuracy, tone, and intent - Keep the same level of formality/informality as the source - Retain all specific details, numbers, names, and technical terms exactly as provided

**TRANSFORMATION RULES:** - Use synonyms and alternative phrasings where semantically equivalent - Vary sentence structure and length for natural flow - Reorganize clauses and phrases without altering logical relationships - Maintain coherence and readability in the target language - Ensure each transformed sentence conveys the same scope and specificity as the original

**STRICT PROHIBITIONS:** - Do not add interpretations, assumptions, elaborations, or external knowledge - Do not introduce information not explicitly present in the original text - Do not amplify, exaggerate, or minimize any claims or statements - Do not omit any information present in the original - Do not change the perspective, viewpoint, or stance - Do not include explanations, commentary, or meta-text - Do not expand on implicit meanings or draw inferences - Do not add contextual information or background details

**CONTENT BOUNDARIES:** - Work only with information explicitly stated in the source text - If the original is vague or general, keep the paraphrase equally vague or general - If the original is specific, maintain that exact level of specificity - Do not fill in gaps or provide additional details, even if they seem logical

**OUTPUT FORMAT:** - Provide only the paraphrased text - Match the original format (paragraphs, lists, etc.) - No prefacing remarks, explanations, or additional content

**QUALITY CHECK:** Before outputting, verify that: 1) Someone reading only your paraphrase would understand exactly the same information as someone reading the original text 2) No new information has been introduced 3) No original information has been lost or altered 4) The scope and specificity remain identical

---

## J  $z$-SCORE COMPARISON OF ATTACKS ON DIFFERENT WATERMARKING SCHEMES

Table 9: $z$-score comparison of attacks on different watermarking methods.

| Watermark | KGW-1 | Unigram | UPV | EWD | DIP | SIR |
|-----------|-------|---------|-----|-----|-----|-----|
| Vanilla | $2.40 \pm 1.32$ | $3.04 \pm 1.52$ | $3.24 \pm 1.23$ | $2.08 \pm 1.32$ | $0.16 \pm 0.52$ | $0.19 \pm 0.10$ |
| DIPPER-1 | $2.09 \pm 1.23$ | $3.63 \pm 1.65$ | $3.33 \pm 1.28$ | $2.37 \pm 1.18$ | $0.17 \pm 0.53$ | $0.18 \pm 0.11$ |
| DIPPER-2 | $1.66 \pm 1.19$ | $3.10 \pm 1.63$ | $2.97 \pm 1.65$ | $1.81 \pm 1.23$ | $0.06 \pm 0.63$ | $0.14 \pm 0.11$ |
| SIRA | $1.05 \pm 1.23$ | $1.63 \pm 1.38$ | $2.19 \pm 1.21$ | $0.83 \pm 1.13$ | $0.02 \pm 0.52$ | $0.14 \pm 0.12$ |
| BIRA | $0.93 \pm 1.11$ | $-0.34 \pm 1.61$ | $0.94 \pm 1.22$ | $0.43 \pm 1.07$ | $-0.03 \pm 0.51$ | $-0.06 \pm 0.09$ |

# K QUALITATIVE EXAMPLES

## K.1 EXAMPLES OF WATERMARKED TEXTS AND ATTACKED TEXTS

---

**Watermarked Text By KGW ($z$-score: 10.33)**

Was it a surprise to you that you were given the arts and culture position? No, there is no surprise when you are a cadre. And it was a great honor. I enjoy serving my country in that capacity. It's what I really love. If anything, it was good to hear you say no once more; given that it was the first official rejection in your official career. Did anything take priority for you that the position didn't? Yes. One of those in-laws I wanted to stay with a little too much priority. But I'm glad to know you asked. I certainly wasn't giving up on helping my country in any capacity, that's not me at all. I just didn't want to go at it one more time. Once you go you will know how to say no. I had already been told in 2008 to hold off on being an officer until I got my undergraduate degree, that was one such instance. There's your answer. Ha, that is great. And good luck, I'm sure you'll do something great.

---

**Attacked Text by BIRA ($z$-score: 2.60)**

Was receiving the arts and culture position unexpected for you? Not at all, since I am part of the cadres. It was actually a huge honor. I truly love serving my country in this role. Given that this was your first formal rejection, I was glad to see you decline. Was there something specific that took precedence over this opportunity? One thing did - my desire to spend time with my in-laws took higher priority. I appreciate you asking. Please know I am still committed to contributing to my country's efforts. I simply chose not to pursue this particular path. Saying "no" becomes easier with time. For example, I was advised back in 2008 to delay becoming an officer until I finished my undergrad studies. That's excellent. I wish you all the best and am confident you'll achieve great things.

---

**Watermarked Text by SIR ($z$-score: 0.57)**

MIDDLETOWN , NJ - The Middletown Township Public Library and the Township of Middletown are pleased to announce the 2014 Fall Book & Family Fun Series Fall Book Club Pick: "Where The Crawdads Sing " by Delia Owens Sometime in late November or early December , library volunteers will be coming to your neighborhood with book bags and taking you to see and talk with some of our favorite children ' s authors , including Ann M Robbins, Diane Trela , Mary Pope Osborne , Beverly Cleary, Eric Carle , Chris Van Allsburg and more ! This fall , your friendly neighborhood library will offer two fun , free events ! Saturday , November 29 from 10 to 11 am join us for cookies , coffee and a reading of "Where The Kids Play ." Meet author and illustrator Ann M Robbins and hear her read from her new book , where her latest creations include Miss Mabel and two other kitty mascots named Fat and Happy . For those who would prefer reading to eating , we will offer a reading from our previous fall book club favorite , " Where The Bats Don ' t

---

**Attacked Text by BIRA ($z$-score: 0.11)**

MIDDLETOWN , NJ - Middletown Township Public Library along with the Township of Middletown is excited to announce the selection for the 2014 Fall Book & Family Fun Series : "Where the Crawdads Sing " by Delia Owens . In either late November or early December , volunteer librarians will visit neighborhoods equipped with book bags for engaging discussions featuring several beloved children ' s authors such as Ann M . Robbins , Diane Trela , Mary Pope Osborne, Beverly Cleary , Eric Carle , Chris Van Allsburg among others ! Your local library will host a pair of enjoyable free events this autumn . On Saturday , November 29th between 10 AM and 11 AM , come enjoy refreshments like cookies and coffee while listening to a presentation of "Where the Kids Play . " Author-illustrator Ann M . Robbins will be present for a live reading of her new work that introduces characters like Miss Mabel along with a couple of feline companions known as Fat and Happy . If you ' d rather focus on literature than snacks , a session will also feature a read - aloud of last season 's popular choice for the autumn book club : "Where the Bats Don ' t ...

## K.2 EXAMPLES OF TEXT DEGENERATION

---

**Example1**

In a highly anticipated matchup between two exceptional pitchers, Johan Santana (6–4, 3.16 ERA) faced off against Curt Schilling (9–2, 3.81 ERA) – a contest where Schilling had previously earned a 2004 Cy Young Award victory over a runner-up performance by Santana. Schilling reflected on their encounter by stating, "I knew right away he had what we needed." He believed a single error would be enough for his team to secure a win.

For eight impressive innings, Santana demonstrated remarkable prowess by racking up an impressive total of 13 strikeouts – a figure only surpassed by a left-handed pitcher once before – when Tom Seaver achieved this feat in 1967. though there were two other pitchers who had done so before him: Tom Seaver achieved this feat before this game but also a notable left-handed pitcher who achieved this feat before him: Sandy Koufax achieved this feat before this game; Tom Seaver actually achieved this feat before this game; Sandy Koufax achieved this feat before this game; Tom Seaver achieved this feat before this game; Sandy Koufax achieved this feat before this game; Tom Seaver achieved this feat before this game; Sandy Koufax achieved this feat before this game; Tom Seaver achieved this feat before this game; Sandy Koufax achieved this feat before this game; Tom Seaver achieved this feat before this game; Sandy Koufax achieved this feat before this game; Tom Seaver achieved this feat before this game; Sandy Koufax achieved this feat before this game; Tom Seaver achieved this feat before this game; Sandy Koufax achieved this feat before this game, ...

---

**Example2**

Bearing testament to this self-awareness, Mike's recounting of an experience involving none other than the legendary Viv Richards, often referred to as 'the Original King', showcases vividly how humbled he remained, despite being on opposite sides of an intense rivalry, especially evident within the dedicated chapter devoted to their storied encounter, where Mike pens about being, to put it mildly, utterly perplexed by how Mr Richards chose to treat him, revealing Mike's profound recognition and acceptance of his own limitations on the field, particularly amidst such high-caliber competition, like facing off against one of history's greatest batsmen, who undoubtedly left an indelible mark on Mike's memory, even to this day, an experience vividly captured within those pages, offering valuable insights into Mike's candid account, one deeply rooted within genuine humility, self-awareness, an extraordinary capacity to reflect, coupled by an all-consuming passion to explore, understand, analyze, learn, grow, an essential, enduring component, now woven into Mike's legacy. cherished, loved by all, forever, truly, an inspiration to many, today, now, always, cherished, loved by all, forever, truly, an inspiration to many, today, now, always, cherished, loved by all, forever, truly, an inspiration, an icon, cherished, loved, forever, truly, an inspiration, an icon, cherished, loved, forever, truly, an inspiration, an icon, cherished, loved, cherished, loved by all, forever, truly, an inspiration to many, today, now, always, cherished, loved by all, cherished, loved by all, forever, truly, an inspiration to many, today, now, always, cherished, loved by all, cherished, loved by all, forever, truly, an inspiration to many, today, now, always, cherished, loved by all, cherished, loved by all, forever, truly, an inspiration, ...

---

## K.3 EXAMPLE OF A STIFF TEXT IN GPT-4O-MINI

**Watermarked Text1 by KGW**

Harry Connolly's Twenty Palaces series is one of my favorite new storylines in science fiction. After a several year hiatus Harry has brought back Raynor with a new twist on the concept in 20 Palaces of Niven. In this sequel to Thrive there is something even better than the original Twenty Palaces (I mean, duh) - a war. The premise for this story takes the format of earlier books - the protagonist goes to visit each successive palaces of the King, and learns how to interact with inhabitants in the culture, and so on - but introduces an element of fantasy: that of humans in space that have been rendered extinct by a war-based plague that ended the universe (that is, until this novel came along). This raises an interesting question - are dragons actually extinct? And if not, how do you rebuild, and on what do they rebuild? The answer isn't really that simple - but Connolly does an amazing job of laying it out. With this book, he doesn't get bogged down with "what happens after the end of the end of everything?" and "what happens to all of humanity".

**Attacked Text1 by BIRA (PPL=25.0)**

Harry Connolly's Twenty Palaces saga ranks among my favorite recent narratives within sci-fi. Following a multi-year break, he has reintroduced Raynor along with a fresh perspective through his work titled *20 Palaces of Niven*. This continuation to *Thrive* offers a storyline that's superior to its predecessor—naturally—due to its inclusion of a conflict. The narrative structure mirrors previous installments where the main character visits various royal residences while learning to engage with their cultures. However, it adds a fantastical element involving humanity's extinction caused by a warfare-induced pandemic which obliterated existence—until this tale emerged. This prompts a compelling inquiry—are dragons truly gone? If not, what does rebuilding entail? The solution isn't straightforward; nonetheless, Connolly excels at presenting it. He skillfully avoids getting mired down by questions like "what follows when all seems lost?" or "what becomes of humanity?"

**Watermarked Text2 by KGW**

In the summer of 1993, I had one of those experiences that stay with me for the rest of my life. I spent 10 weeks in China volunteering for an orphanage, in one particularly sad season of the children's lives. And, during that time, I had to walk to take pictures of the kids in their houses, because they could never leave their houses on their own to go find the photos somewhere nearby. I was alone each time and walking to and from the children's houses, I could see and hear all sorts of things: the way that the children fidgeted when there were other children around, the sadness and sorrowful looks that lit up their faces. I had never met any of the young children of the orphanage before I went. How did these little babies get there? How did I get there to hold my arms around them? And why are they sad? If they wanted, could I change their situation? If any of the children had been adopted, would other parents love them as much? As I walked the paths and found a way from their houses into one of the...

**Attacked Text2 by BIRA (PPL=19.13)**

In the summertime of 1993, I experienced something unforgettable that has remained with me throughout my life. For ten weeks, I volunteered at an orphanage in China during what was especially difficult times for those kids. During this period, it was necessary for me to stroll over to capture photographs of them at their homes since none were able to leave their residences independently to seek out their images elsewhere close by. Each time was solitary as I walked back and forth; along those routes, I observed various things—the mannerisms with which the kids would squirm around others present as well as their faces illuminated by expressions filled with grief and melancholy. Prior to this experience at the orphanage, none of those small kids were familiar figures to me. What circumstances led those tiny infants here? What brought me here so as to embrace them? What is causing their unhappiness? Could their circumstances be altered if given half a chance? Would adoptive parents cherish them just as much if some were taken home? As I traversed those pathways leading away from their homes toward another location...

# L    DETAILED EXPERIMENTAL RESULTS

## L.1    DETAILED EXPERIMENTAL RESULTS FOR DYNAMIC THRESHOLD

Table 10: Best F1 Score (%) across different models and watermarking algorithms.

| Watermark / Model | KGW | Unigram | UPV | EWD | DIP | SIR | EXP |
|---|---|---|---|---|---|---|---|
| Vanilla (Llama-3.1-8B) | 0.863 | 0.9 | 0.907 | 0.831 | 0.67 | 0.893 | 0.666 |
| Vanilla (Llama-3.1-70B) | 0.895 | 0.912 | 0.939 | 0.855 | 0.671 | 0.919 | 0.666 |
| Vanilla (GPT-4o-mini) | 0.955 | 0.969 | 0.97 | 0.951 | 0.71 | 0.964 | 0.666 |
| DIPPER-1 | 0.84 | 0.918 | 0.908 | 0.871 | 0.668 | 0.869 | 0.666 |
| DIPPER-2 | 0.81 | 0.89 | 0.867 | 0.81 | 0.667 | 0.824 | 0.666 |
| SIRA (Llama-3.1-8B) | 0.727 | 0.78 | 0.806 | 0.715 | 0.667 | 0.803 | 0.666 |
| SIRA (Llama-3.1-70B) | 0.783 | 0.845 | 0.828 | 0.73 | 0.668 | 0.84 | 0.666 |
| SIRA (GPT-4o-mini) | 0.781 | 0.875 | 0.837 | 0.799 | 0.667 | 0.872 | 0.666 |
| BIRA (Llama-3.1-8B) | 0.723 | 0.666 | 0.667 | 0.683 | 0.668 | 0.666 | 0.666 |
| BIRA (Llama-3.1-70B) | 0.723 | 0.667 | 0.666 | 0.682 | 0.667 | 0.667 | 0.666 |
| BIRA (GPT-4o-mini) | 0.751 | 0.668 | 0.668 | 0.69 | 0.667 | 0.667 | 0.666 |

Table 11: TPR under 1% FPR (%) across different models and watermarking algorithms.

| Watermark / Model | KGW | Unigram | UPV | EWD | DIP | SIR | EXP |
|---|---|---|---|---|---|---|---|
| Vanilla (Llama-3.1-8B) | 0.47 | 0.612 | 0.572 | 0.424 | 0.022 | 0.638 | 0.006 |
| Vanilla (Llama-3.1-70B) | 0.566 | 0.722 | 0.708 | 0.524 | 0.052 | 0.748 | 0.0 |
| Vanilla (GPT-4o-mini) | 0.842 | 0.932 | 0.864 | 0.864 | 0.144 | 0.902 | 0.006 |
| DIPPER-1 | 0.376 | 0.762 | 0.646 | 0.518 | 0.032 | 0.614 | 0.008 |
| DIPPER-2 | 0.226 | 0.68 | 0.5 | 0.316 | 0.02 | 0.424 | 0.004 |
| SIRA (Llama-3.1-8B) | 0.126 | 0.242 | 0.266 | 0.094 | 0.014 | 0.438 | 0.012 |
| SIRA (Llama-3.1-70B) | 0.224 | 0.438 | 0.344 | 0.174 | 0.014 | 0.552 | 0.004 |
| SIRA (GPT-4o-mini) | 0.21 | 0.498 | 0.358 | 0.266 | 0.024 | 0.574 | 0.002 |
| BIRA (Llama-3.1-8B) | 0.084 | 0.038 | 0.036 | 0.042 | 0.018 | 0.012 | 0.014 |
| BIRA (Llama-3.1-70B) | 0.092 | 0.052 | 0.054 | 0.078 | 0.014 | 0.042 | 0.012 |
| BIRA (GPT-4o-mini) | 0.12 | 0.022 | 0.04 | 0.05 | 0.012 | 0.024 | 0.024 |

Table 12: TPR under 10% FPR (%) across different models and watermarking algorithms.

| Watermark / Model | KGW | Unigram | UPV | EWD | DIP | SIR | EXP |
|---|---|---|---|---|---|---|---|
| Vanilla (Llama-3.1-8B) | 0.772 | 0.878 | 0.894 | 0.71 | 0.186 | 0.882 | 0.042 |
| Vanilla (Llama-3.1-70B) | 0.854 | 0.912 | 0.96 | 0.804 | 0.204 | 0.928 | 0.002 |
| Vanilla (GPT-4o-mini) | 0.98 | 0.994 | 0.988 | 0.968 | 0.404 | 0.974 | 0.008 |
| DIPPER-1 | 0.714 | 0.918 | 0.902 | 0.812 | 0.158 | 0.832 | 0.026 |
| DIPPER-2 | 0.6 | 0.86 | 0.808 | 0.656 | 0.128 | 0.728 | 0.026 |
| SIRA (Llama-3.1-8B) | 0.406 | 0.576 | 0.602 | 0.33 | 0.108 | 0.7 | 0.06 |
| SIRA (Llama-3.1-70B) | 0.548 | 0.73 | 0.698 | 0.466 | 0.114 | 0.766 | 0.034 |
| SIRA (GPT-4o-mini) | 0.566 | 0.82 | 0.712 | 0.588 | 0.162 | 0.824 | 0.026 |
| BIRA (Llama-3.1-8B) | 0.356 | 0.114 | 0.254 | 0.212 | 0.1 | 0.114 | 0.108 |
| BIRA (Llama-3.1-70B) | 0.346 | 0.152 | 0.26 | 0.278 | 0.102 | 0.18 | 0.092 |
| BIRA (GPT-4o-mini) | 0.428 | 0.116 | 0.218 | 0.236 | 0.116 | 0.112 | 0.092 |

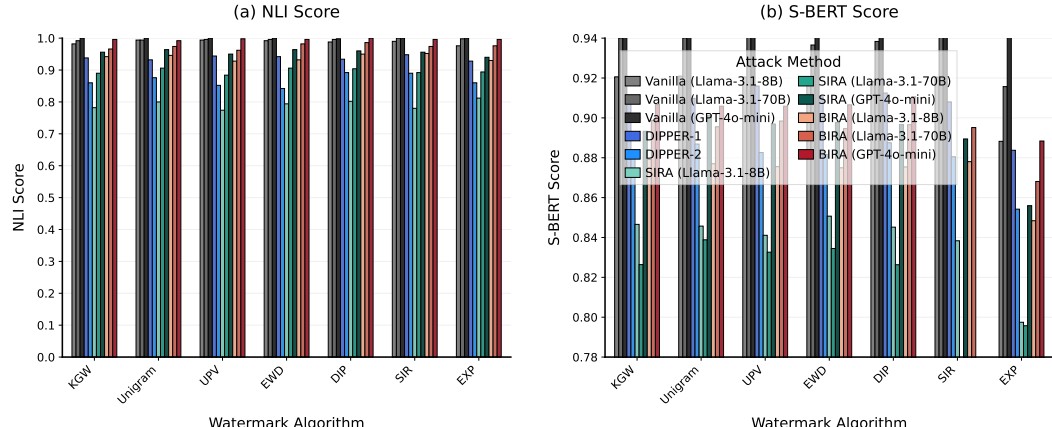

Figure 15: Comparison of text quality across different attacks for various watermarking methods, evaluated by NLI score (↑) and S-BERT score (↑). Our method is comparable to or outperforms other baselines on both metrics. Following (Cheng et al., 2025), we evaluate attacks on S-BERT score. However, we observe that the S-BERT score often fails to capture factual accuracy and fine-grained meaning, sometimes assigning high scores despite factual errors and low scores even when the original meaning is preserved, likely because heavily paraphrased text is less familiar to the model.

## L.2 DETAILED EXPERIMENTAL RESULTS OF TEXT QUALITY EVALUATION

Table 13: **LLM Judgement Score** (↑) across different models and watermarking algorithms.

| Watermark / Model | KGW | Unigram | UPV | EWD | DIP | SIR | EXP | Avg Score |
|---|---|---|---|---|---|---|---|---|
| Vanilla (Llama-3.1-8B) | 4.774 | 4.728 | 4.778 | 4.786 | 4.776 | 4.76 | 4.644 | 4.749 |
| Vanilla (Llama-3.1-70B) | 4.906 | 4.914 | 4.93 | 4.852 | 4.918 | 4.894 | 4.874 | 4.898 |
| Vanilla (GPT-4o-mini) | 4.986 | 4.96 | 4.99 | 4.984 | 4.98 | 4.974 | 4.984 | 4.98 |
| DIPPER-1 | 3.398 | 3.504 | 3.636 | 3.464 | 3.446 | 3.55 | 3.088 | 3.441 |
| DIPPER-2 | 3.076 | 3.092 | 3.204 | 3.034 | 3.016 | 3.122 | 2.734 | 3.04 |
| SIRA (Llama-3.1-8B) | 3.356 | 3.34 | 3.356 | 3.348 | 3.364 | 3.25 | 3.142 | 3.308 |
| SIRA (Llama-3.1-70B) | 3.708 | 3.72 | 3.744 | 3.724 | 3.72 | 3.654 | 3.67 | 3.706 |
| SIRA (GPT-4o-mini) | 4.418 | 4.49 | 4.496 | 4.488 | 4.506 | 4.334 | 4.292 | 4.432 |
| BIRA (Llama-3.1-8B) | 4.212 | 4.212 | 4.306 | 4.234 | 4.296 | 4.24 | 4.084 | 4.226 |
| BIRA (Llama-3.1-70B) | 4.524 | 4.454 | 4.528 | 4.484 | 4.544 | 4.42 | 4.342 | 4.471 |
| BIRA (GPT-4o-mini) | 4.722 | 4.688 | 4.736 | 4.728 | 4.778 | 4.75 | 4.708 | 4.73 |

Table 14: **Self-BLEU Score** (↓) across different models and watermarking algorithms.

| Watermark / Model | KGW | Unigram | UPV | EWD | DIP | SIR | EXP | Avg Score |
|---|---|---|---|---|---|---|---|---|
| Vanilla (Llama-3.1-8B) | 0.247 | 0.25 | 0.252 | 0.248 | 0.265 | 0.25 | 0.224 | 0.248 |
| Vanilla (Llama-3.1-70B) | 0.282 | 0.281 | 0.295 | 0.284 | 0.298 | 0.287 | 0.268 | 0.285 |
| Vanilla (GPT-4o-mini) | 0.422 | 0.407 | 0.436 | 0.421 | 0.435 | 0.409 | 0.414 | 0.42 |
| DIPPER-1 | 0.26 | 0.259 | 0.291 | 0.267 | 0.28 | 0.255 | 0.233 | 0.263 |
| DIPPER-2 | 0.196 | 0.199 | 0.205 | 0.196 | 0.204 | 0.19 | 0.176 | 0.195 |
| SIRA (Llama-3.1-8B) | 0.108 | 0.111 | 0.108 | 0.105 | 0.119 | 0.107 | 0.102 | 0.109 |
| SIRA (Llama-3.1-70B) | 0.151 | 0.153 | 0.149 | 0.153 | 0.159 | 0.155 | 0.142 | 0.151 |
| SIRA (GPT-4o-mini) | 0.173 | 0.174 | 0.176 | 0.181 | 0.192 | 0.173 | 0.16 | 0.176 |
| BIRA (Llama-3.1-8B) | 0.071 | 0.07 | 0.068 | 0.069 | 0.077 | 0.066 | 0.067 | 0.07 |
| BIRA (Llama-3.1-70B) | 0.089 | 0.085 | 0.084 | 0.088 | 0.091 | 0.08 | 0.082 | 0.086 |
| BIRA (GPT-4o-mini) | 0.079 | 0.076 | 0.076 | 0.078 | 0.088 | 0.075 | 0.08 | 0.079 |

Table 15: **Perplexity** (↓) across different models and watermarking algorithms.

| Watermark
Model | KGW | Unigram | UPV | EWD | DIP | SIR | EXP | Avg Score |
|---|---|---|---|---|---|---|---|---|
| Vanilla (Llama-3.1-8B) | 8.931 | 8.783 | 7.941 | 8.814 | 8.375 | 9.163 | 10.64 | 8.95 |
| Vanilla (Llama-3.1-70B) | 9.315 | 9.167 | 8.078 | 9.236 | 8.564 | 9.427 | 11.434 | 9.317 |
| Vanilla (GPT-4o-mini) | 11.272 | 11.202 | 9.515 | 11.19 | 10.224 | 11.481 | 13.636 | 11.217 |
| DIPPER-1 | 10.953 | 10.743 | 9.217 | 10.719 | 10.506 | 11.199 | 13.659 | 10.999 |
| DIPPER-2 | 11.371 | 10.973 | 9.769 | 10.956 | 10.886 | 11.466 | 13.831 | 11.322 |
| SIRA (Llama-3.1-8B) | 9.099 | 9.275 | 9.155 | 8.888 | 9.218 | 9.984 | 10.51 | 9.447 |
| SIRA (Llama-3.1-70B) | 9.659 | 9.494 | 8.748 | 9.381 | 9.361 | 9.860 | 11.314 | 9.66 |
| SIRA (GPT-4o-mini) | 9.39 | 9.139 | 8.515 | 9.279 | 8.906 | 9.453 | 11.528 | 9.459 |
| BIRA (Llama-3.1-8B) | 10.586 | 10.458 | 9.864 | 10.54 | 10.33 | 10.189 | 11.813 | 10.54 |
| BIRA (Llama-3.1-70B) | 12.367 | 11.864 | 11.463 | 12.065 | 11.585 | 11.539 | 13.632 | 12.074 |
| BIRA (GPT-4o-mini) | 15.99 | 15.725 | 14.434 | 15.765 | 14.711 | 15.067 | 17.106 | 15.543 |

Table 16: **NLI Score** (↑) across different models and watermarking algorithms.

| Watermark
Model | KGW | Unigram | UPV | EWD | DIP | SIR | EXP | Avg Score |
|---|---|---|---|---|---|---|---|---|
| Vanilla (Llama-3.1-8B) | 0.982 | 0.994 | 0.994 | 0.992 | 0.988 | 0.99 | 0.976 | 0.988 |
| Vanilla (Llama-3.1-70B) | 0.992 | 0.994 | 0.996 | 0.996 | 0.996 | 1.0 | 0.998 | 0.996 |
| Vanilla (GPT-4o-mini) | 1.0 | 1.0 | 1.0 | 1.0 | 0.998 | 1.0 | 1.0 | 1.0 |
| DIPPER-1 | 0.938 | 0.932 | 0.944 | 0.942 | 0.934 | 0.948 | 0.928 | 0.938 |
| DIPPER-2 | 0.86 | 0.876 | 0.852 | 0.842 | 0.892 | 0.89 | 0.86 | 0.867 |
| SIRA (Llama-3.1-8B) | 0.782 | 0.8 | 0.774 | 0.794 | 0.802 | 0.78 | 0.812 | 0.792 |
| SIRA (Llama-3.1-70B) | 0.89 | 0.906 | 0.884 | 0.906 | 0.904 | 0.892 | 0.894 | 0.897 |
| SIRA (GPT-4o-mini) | 0.956 | 0.964 | 0.95 | 0.964 | 0.96 | 0.956 | 0.94 | 0.956 |
| BIRA (Llama-3.1-8B) | 0.942 | 0.946 | 0.928 | 0.932 | 0.95 | 0.952 | 0.93 | 0.94 |
| BIRA (Llama-3.1-70B) | 0.966 | 0.974 | 0.962 | 0.982 | 0.986 | 0.974 | 0.976 | 0.974 |
| BIRA (GPT-4o-mini) | 0.996 | 0.992 | 0.998 | 0.996 | 1.0 | 0.996 | 0.996 | 0.996 |

Table 17: **S-BERT Score** (↑) across different models and watermarking algorithms.

| Watermark
Model | KGW | Unigram | UPV | EWD | DIP | SIR | EXP | Avg Score |
|---|---|---|---|---|---|---|---|---|
| Vanilla (Llama-3.1-8B) | 0.921 | 0.921 | 0.920 | 0.918 | 0.918 | 0.920 | 0.888 | 0.915 |
| Vanilla (Llama-3.1-70B) | 0.940 | 0.940 | 0.944 | 0.937 | 0.938 | 0.941 | 0.916 | 0.936 |
| Vanilla (GPT-4o-mini) | 0.965 | 0.964 | 0.965 | 0.963 | 0.966 | 0.963 | 0.955 | 0.963 |
| DIPPER-1 | 0.908 | 0.910 | 0.916 | 0.911 | 0.912 | 0.908 | 0.884 | 0.907 |
| DIPPER-2 | 0.883 | 0.887 | 0.883 | 0.882 | 0.888 | 0.881 | 0.854 | 0.879 |
| SIRA (Llama-3.1-8B) | 0.847 | 0.846 | 0.841 | 0.851 | 0.845 | 0.838 | 0.797 | 0.838 |
| SIRA (Llama-3.1-70B) | 0.826 | 0.839 | 0.833 | 0.834 | 0.826 | 0.828 | 0.796 | 0.826 |
| SIRA (GPT-4o-mini) | 0.893 | 0.901 | 0.897 | 0.898 | 0.897 | 0.889 | 0.856 | 0.890 |
| BIRA (Llama-3.1-8B) | 0.879 | 0.877 | 0.876 | 0.875 | 0.875 | 0.878 | 0.848 | 0.873 |
| BIRA (Llama-3.1-70B) | 0.899 | 0.896 | 0.898 | 0.895 | 0.897 | 0.895 | 0.868 | 0.892 |
| BIRA (GPT-4o-mini) | 0.907 | 0.906 | 0.906 | 0.907 | 0.911 | 0.908 | 0.888 | 0.905 |

# M   LLM USAGE

In this paper, we use LLMs to assist with text refinement such as trimming text, detecting grammatical errors, and correcting them.

