# OpenReview forum: "LLM Watermark Evasion via Bias Inversion"
_ICLR.cc/2026/Conference — Submitted to ICLR 2026_

### Official Review · Reviewer_WgyA · 2025-10-20

**Soundness:** 3
**Presentation:** 4
**Contribution:** 3
**Rating:** 6
**Confidence:** 4

**Summary:**

The authors present a theoretical and empirical study of vulnerabilities in current watermarking schemes for LLMs. They introduce BIRA, a query-free, model-agnostic method that systematically removes watermarks by applying a negative logit bias to tokens with high self-information, thereby suppressing those most likely to carry the watermark signal. They further provided theoretical analysis showing that reducing the average probability of green tokens by a small margin $\delta$ causes the watermark detection probability to decay exponentially as $\Pr[D(\tilde{y}, W_k)=1] \le e^{-N\delta^2/2}$. They ran extensive experiments across seven watermarking algorithms and multiple LLMs (Llama-3.1, GPT-4o-mini) and show $>99%$ evasion success while maintaining high semantic fidelity and fluency.

**Strengths:**

- They showed that watermark detection probability decays exponentially with even small reductions in green-token bias, certifying their novelty.
- The proposed approach is both query-free and model-agnostic, requiring no secret knowledge, fine-tuning, or training.
- Their comprehensive empirical validation across seven watermarking methods and multiple LLM families (Llama, GPT-4o) is enough to drive home this contribution.
- The writing and presentation are clear, structured, and polished.

**Weaknesses:**

- It is well-known in literature that paraphrasing attacks mostly succeeds in evading detection. So, proposing yet another attack seems like confirming what has already been confirmed.
- Although the self-information heuristic for selecting proxy tokens is intuitive, it remains heuristic; providing a theoretical or empirical justification (e.g., correlation with actual green-token distributions) would reinforce confidence in its generality.
- No defenses were proposed nor evaluated in this case.
- The paper would benefit from an included discussion and evaluation of adaptive paraphrasing attacks [1, 2], which currently achieve the best evasion–fidelity trade-off. Explicitly contrasting BIRA’s query-free design with these adaptive attackers would greatly strengthen the motivation and highlight its advantage of achieving similar success without watermark or detector knowledge.

**References**

[1] Jovanović N, Staab R, Vechev M. Watermark stealing in large language models. arXiv preprint arXiv:2402.19361. 2024 Feb 29.

[2] Diaa A, Aremu T, Lukas N. Optimizing Adaptive Attacks against Watermarks for Language Models. arXiv preprint arXiv:2410.02440. 2024 Oct 3.

**Questions:**

Please see weaknesses above

---

> ### Author Response · Authors · 2025-11-20
> **Response to Reviewer WgyA**
>
> #### Thank you for taking the time to review and engage with our paper. We have carefully considered your comments and provide detailed responses below. We have also added the corresponding experimental results comparing against the adaptive attack in Appendix G of the revised manuscript, marked in pink.
>
> > W1. It is well-known in literature that paraphrasing attacks mostly succeed in evading detection. So, proposing yet another attack seems like confirming what has already been confirmed.
> > W4. The paper would benefit from an included discussion and evaluation of adaptive paraphrasing attacks [1, 2], which currently achieve the best evasion–fidelity trade-off. Explicitly contrasting BIRA’s query-free design with these adaptive attackers would greatly strengthen the motivation and highlight its advantage of achieving similar success without watermark or detector knowledge.
>
> #### While paraphrasing attacks [1,2] are common in recent literature, our method differs in both threat model and methodology. First, we operate under a more realistic threat model. Second, our attack is directly motivated by a theoretical characterization of a fundamental vulnerability in current watermarking schemes (Theorem 2).
>
> #### Existing attacks rely on assumptions that are often impractical in real-world settings:
> - #### [1] is a query-based attack that uses repeated, carefully designed prompts to recover the green token set from the watermarked LLM. This is computationally expensive and has limited practicality in rate-limited API settings. Moreover, [1] is specifically tailored to KGW watermarks and assumes knowledge of the hash function that defines the green list.
> - #### [2] is a query-free attack but assumes knowledge of the watermarking scheme and considers an adaptive scenario in which the adversary fine-tunes a specialized paraphrasing model for the target watermark, relying on transferability to unseen schemes.
>
> #### By contrast, our method (BIRA) is **query-free, scheme-agnostic, and training-free**. It does not query the watermarked LLM, does not assume any knowledge of the watermarking scheme, and does not require training an additional paraphrasing model. Theorem 2 shows that even a small reduction in the green-token sampling probability, on average over the sequence, is sufficient to make the text statistically undetectable. Motivated by this insight, instead of accurately identifying the green list as in [1] or training a paraphrasing model as in [2], BIRA approximates a proxy green set using token self-information and applies a negative logit bias to these tokens during rewriting. Under this more practical threat model, BIRA achieves high evasion rates while preserving semantic fidelity across seven recent watermarking methods.
>
> #### Additionally, while our main evaluation focuses on query-free attacks without knowledge of the watermarking scheme, we also compare against the adaptive attack [2] in Appendix G. Interestingly, BIRA and SIRA achieve higher attack success than [2] (Table 8 and Figure 13), whereas [2] attains higher semantic fidelity than BIRA and SIRA in terms of LLM judgment score (Figure 14). These results indicate that BIRA is a substantially stronger attack in terms of evasion, with a modest trade-off in semantic fidelity.
>
> >W2. Although the self-information heuristic for selecting proxy tokens is intuitive, it remains heuristic; providing a theoretical or empirical justification (e.g., correlation with actual green-token distributions) would reinforce confidence in its generality.
> #### We provide empirical validation of this correlation in Section 5.3, where we conduct an ablation study comparing self-information-guided token selection against a random selection baseline using the same selection ratio.
> #### Figure 5 demonstrates that self-information-guided selection consistently outperforms random selection, indicating that it more effectively targets green tokens. Given that watermarked text is dominated by green tokens, even random selection naturally captures many of them. Therefore, the additional performance gain from our method confirms that high self-information is significantly more correlated with the watermark signal than a random token selection.
>
>
> >W3. No defenses were proposed nor evaluated in this case.
>
> #### We acknowledge that our study does not propose or evaluate defense methods. We view designing new watermarking schemes that explicitly account for the vulnerabilities characterized by Theorem 2 as an important direction for future work, and we will make this point explicit in the revised manuscript.
>
> #### [1] Jovanovic et al. Watermark Stealing in Large Language Models. ICML 2024.
> #### [2] Diaa et al. Optimizing Adaptive Attacks against Watermarks for Language Models. ICLR 2025 workshop

---

> > ### Comment · Reviewer_WgyA · 2025-11-25
> >
> > Thank you for providing the response. I now see the positioning more clearly, and although the end goal (effective watermark evasion) remains the same, the assumptions and attack mechanisms differ meaningfully. However, I stand with my notion that evasion attacks tend to be incremental, given the fundamental difficulty of designing provably strong watermarks (“Watermarks in the Sand”, ICML 2024). Still, new, practically motivated attacks like BIRA are crucial for benchmarking future watermark robustness. I’ll be keeping my score as I remain personally more biased toward constructive defense research than attack-oriented work. Kudos and goodluck to the authors!

---

> > > ### Author Response · Authors · 2025-11-25
> > > **Response to Reviewer WgyA**
> > >
> > > Thank you for your constructive and thoughtful feedback. It has helped us strengthen the paper, and we appreciate your engagement and endorsement during the rebuttal process.

---

### Official Review · Reviewer_MsV7 · 2025-10-21

**Soundness:** 3
**Presentation:** 3
**Contribution:** 2
**Rating:** 6
**Confidence:** 5

**Summary:**

This paper presents BIRA, a method for evading LLM watermarks through adversarial paraphrasing. The authors first establish theoretical foundations showing that watermark detection reduces to a threshold test on green token rates, and that suppressing green token probabilities leads to exponential decay in detection probability. Building on this insight, BIRA identifies likely watermarked tokens using self-information, then applies negative logit biases during LLM-based paraphrasing to systematically reduce green token generation. Experiments across seven watermarking schemes demonstrate that BIRA achieves over 99% attack success rates while maintaining semantic fidelity, significantly outperforming existing query-free attacks like DIPPER and SIRA.

**Strengths:**

1. Unlike previous methods that rely on coarse masking-and-rewriting strategies, BIRA proposes a more fine-grained paraphrasing mechanism by applying token-level negative biases at every generation step.
2. The method demonstrates consistently high attack success rates across multiple watermarking schemes and different language models (Llama-3.1-8B/70B, GPT-4o-mini).
3. The method is straightforward to understand and implement. The two-phase approach (proxy green set construction + bias-inversion rewriting) is intuitive, and the use of self-information for identifying likely watermarked tokens is well-motivated.

**Weaknesses:**

- While the paper claims to preserve semantic fidelity, Table 2 and Table 11 reveal that perplexity increases significantly with stronger attacks (from 9.17 at β=0 to 14.10 at β=-9.0 for Llama models, and notably higher for GPT-4o-mini at 15.5-17.1). The authors acknowledge in Appendix F.3 that GPT-4o-mini produces "stiff text" with "unconventional vocabulary," but this critical limitation receives minimal discussion in the main paper. The trade-off between attack efficacy and text naturalness deserves deeper analysis: At what perplexity threshold does the attacked text become noticeably unnatural to human readers? Could this quality degradation itself serve as a detection signal? The paper would benefit from human evaluation studies assessing whether the perplexity increase translates to perceptible quality loss in real-world scenarios.

- The paper exclusively evaluates BIRA against token-level watermarking schemes where the watermark signal is embedded through biased token sampling. However, recent work has explored alternative watermarking paradigms that operate at different granularities, such as sentence-level watermarks, semantic watermarks that constrain meaning rather than lexical choices, or structural watermarks embedded in discourse patterns. Since BIRA's core mechanism relies on suppressing token-level green tokens identified through self-information, it remains unclear whether this approach generalizes to non-token-based watermarking methods. The authors should discuss these limitations and provide theoretical or empirical analysis on whether BIRA's principles (suppressing watermark signals during rewriting) extend to other watermarking paradigms, or if fundamentally different attack strategies would be required.

**Questions:**

- Could you provide more detailed analysis on the perplexity increase? Specifically: (a) Have you conducted human evaluation to assess whether the perplexity degradation (especially for GPT-4o-mini with PPL ~15-17) results in noticeably unnatural text that human readers can detect? (b) Could detectors exploit this quality degradation as a secondary signal? For instance, could a two-stage detector first check for watermark absence, then flag texts with suspicious perplexity patterns as potentially attacked? (c) What is the perplexity distribution of genuinely human-written text versus BIRA-attacked text, and is there sufficient overlap to avoid detection?

- How would BIRA perform against watermarking schemes that don't rely on token-level biases? For example: (a) Sentence-level watermarks that embed signals across entire sentences rather than individual tokens? (b) Semantic watermarks that constrain the semantic space rather than lexical choices (c) Have you considered evaluating against or discussing these alternative watermarking paradigms to clarify the scope of BIRA's applicability?

---

> ### Author Response · Authors · 2025-11-20
> **Response to Reviewer MsV7 (Part 1/2)**
>
> #### Thank you for taking the time to review our work and for providing insightful feedback. We have carefully considered your comments and provide detailed responses below. In the revised manuscript, we have also added a human evaluation of the attacked text with increased PPL (Appendix F) and an extension of our theorem to semantic, sentence-level watermarking, along with additional experiments and analyses (Appendix E), marked in pink
>
> > W1. While the paper claims to preserve semantic fidelity, Table 2 and Table 11 reveal that perplexity increases significantly with stronger attacks (from 9.17 at β=0 to 14.10 at β=-9.0 for Llama models, and notably higher for GPT-4o-mini at 15.5-17.1). The authors acknowledge in Appendix F.3 that GPT-4o-mini produces "stiff text" with "unconventional vocabulary," but this critical limitation receives minimal discussion in the main paper. Could you provide more detailed analysis on the perplexity increase?  The paper would benefit from human evaluation studies assessing whether the perplexity increase translates to perceptible quality loss in real-world scenarios.
>
> > Q1-(a). Have you conducted human evaluation to assess whether the perplexity degradation (especially for GPT-4o-mini with PPL ~15-17) results in noticeably unnatural text that human readers can detect?
>
> #### We manually inspected BIRA-attacked samples with high perplexity for GPT-4o-mini and found that, despite their elevated PPL, the texts remain grammatical and reasonably readable (see qualitative examples in Appendix K.3). Our human evaluation suggests that this PPL increase may stem from stylistic factors such as overuse of hyphens or particular synonym choices. Crucially, we also observe that human-written text from the C4 dataset can exhibit similarly high PPL, particularly in informal or conversational sentences (Figure 11). This supports the view that PPL is not a perfect proxy for human-perceived quality in real-world scenarios.
>
>
> >Q1-(b-c). (b) Could detectors exploit this quality degradation as a secondary signal? For instance, could a two-stage detector first check for watermark absence, then flag texts with suspicious perplexity patterns as potentially attacked? (c) What is the perplexity distribution of genuinely human-written text versus BIRA-attacked text, and is there sufficient overlap to avoid detection?
>
> #### We observed that perplexity alone is not a sufficiently reliable detection signal because the distributions of natural and attacked text overlap significantly (Appendix F). As illustrated in Figure 12, the PPL distributions of attacked text by BIRA for KGW watermarking and natural text in C4 dataset overlap substantially for Llama-3.1-8B and Llama-3.1-70B, making PPL ineffective for discriminating between them. For GPT-4o-mini, the BIRA distribution is more shifted toward higher PPL, but there is still considerable overlap with natural text. We suspect that part of this shift is caused by the mismatch between GPT-4o-mini’s tokenizer and the auxiliary tokenizer (Llama-3.1-3B) used to evaluate token self-information in our query-free attack setup.
>
> #### Additionally, unlike statistical watermarking, which provides formal guarantees via a calibrated test statistic, PPL-based detection has no such theoretical grounding. Thus, although designing BIRA variants that lower PPL while preserving attack strength is an interesting future work, perplexity remains a heuristic and lacks the robustness needed for a reliable post-hoc detection mechanism.

---

> ### Author Response · Authors · 2025-11-20
> **Response to Reviewer MsV7 (Part 2/2)**
>
> > W2. The paper exclusively evaluates BIRA against token-level watermarking schemes where the watermark signal is embedded through biased token sampling. However, recent work has explored alternative watermarking paradigms that operate at different granularities, such as sentence-level watermarks, semantic watermarks that constrain meaning rather than lexical choices, or structural watermarks embedded in discourse patterns. Since BIRA's core mechanism relies on suppressing token-level green tokens identified through self-information, it remains unclear whether this approach generalizes to non-token-based watermarking methods. The authors should discuss these limitations and provide theoretical or empirical analysis on whether BIRA's principles (suppressing watermark signals during rewriting) extend to other watermarking paradigms, or if fundamentally different attack strategies would be required.
>
> > Q1-1. How would BIRA perform against watermarking schemes that don't rely on token-level biases? (a) Sentence-level watermarks that embed signals across entire sentences rather than individual tokens? (b) Semantic watermarks that constrain the semantic space rather than lexical (c) Have you considered evaluating against or discussing these alternative watermarking paradigms to clarify the scope of BIRA's applicability?
>
> #### We appreciate the reviewer’s insightful feedback. While our method is primarily designed for token-level watermarking schemes, which are both fundamental and currently the dominant paradigm, our theoretical framework naturally extends to recent semantic, sentence-level watermarking methods due to their conceptual similarity to token-based schemes (Appendix E). Specifically, these methods count “valid” (green) sentence pairs in an embedding space in a way that is structurally analogous to counting green tokens in token-level schemes, which suggests that our approach can be extended to this setting.
>
> #### Furthermore, we empirically demonstrate that BIRA is effective at evading the state-of-the-art semantic, sentence-level watermark SimMark [1] (Table 7). Although its evasion rate is slightly lower than in the token-level setting, BIRA still achieves strong evasion performance, indicating that it remains effective against non-token-based watermarking methods.
>
> #### [1] A DabiriAghdam et al. SimMark: A Robust Sentence-Level Similarity-Based Watermarking Algorithm for Large Language Models. EMNLP 2025

---

> ### Author Response · Authors · 2025-11-27
> **A Gentle Reminder**
>
> #### We would like to kindly remind you that the discussion period will end in 6 days. We would greatly appreciate it if you could let us know whether our rebuttal has sufficiently addressed your concerns. Thank you once again for your time and effort.

---

### Official Review · Reviewer_PeB8 · 2025-10-25

**Soundness:** 2
**Presentation:** 3
**Contribution:** 1
**Rating:** 2
**Confidence:** 5

**Summary:**

In this paper, the authors present a bias inversion method for attacking LM watermarking. The method focuses on red-green list styled watermark, in which the watermark is embedded through adding bias to the green list tokens. In the proposed method, we first identify a proxy green token set, then add a negative bias on the logits of the green tokens to erase the watermark.

**Strengths:**

1. The experiments on various LM watermarking show the effectiveness of the proposed method.

**Weaknesses:**

The authors missed an important line of watermarking removal works including [1,2], which are very related to their proposed method. The watermarking attacks introduced in [1,2] are almost the same as the proposed methods, they both focus on a) identify the green list and then b) remove the watermark by adding a negative delta to the green list token logits. Moreover, [2] introduced an unbiased green list and watermarking strength estimating algorithm, which can remove the watermark without hurting the generation quality of the language model. Given the existing work [1,2], the proposed method is not novel and there is no methodology contributions.

[1] Jovanovic et al. Watermark Stealing in Large Language Models. ICML 2024.

[2] Chen et al. De-mark: Watermark Removal in Large Language Models. ICML 2025.

**Questions:**

Is the proposed method superior than [1,2]?

---

> ### Author Response · Authors · 2025-11-20
> **Response to Reviewer PeB8**
>
> #### Thank you for taking the time to review and engage with our paper. We have carefully considered your comments and provide a detailed response below.
> > W1. The authors missed an important line of watermarking removal works including [1,2], which are very related to their proposed method. The watermarking attacks introduced in [1,2] are almost the same as the proposed methods, they both focus on a) identify the green list and then b) remove the watermark by adding a negative delta to the green list token logits. Moreover, [2] introduced an unbiased green list and watermarking strength estimating algorithm, which can remove the watermark without hurting the generation quality of the language model. Given the existing work [1,2], the proposed method is not novel and there is no methodology contributions.
>
>
> #### We respectfully clarify that our method differs from [1,2] in both its threat model and methodology, as discussed in the Related Work section. First, we operate under a more realistic threat model. Second, our attack is directly motivated by a theoretical characterization of a fundamental vulnerability in current watermarking schemes (Theorem 2).
>
> #### Specifically, [1,2] are query-based attacks that use repeated, carefully designed prompts to identify the green token from the watermarked LLM. This is computationally expensive ([2] takes around one hour per sample in our runs) and has limited practicality in rate-limited API settings. Moreover, both methods are specifically tailored to KGW watermarks and rely on specific assumptions: [1] assumes knowledge of the hash function that defines the green list, while [2] requires access to the watermarked LLM’s top-$k$ token probabilities, which is typically infeasible for proprietary black-box systems.
>
> #### In addition, the goal of our work and that of [2] is fundamentally different. [2] treats the watermarked LLM as the attack target, adjusting the watermarked LLM’s probabilities so its behavior approximates the original, unwatermarked model. Our setting instead treats the watermarked text as the target: given a watermarked text, we rewrite it via BIRA to remove the watermark signal.
>
> #### By contrast, our method (BIRA) is query-free and scheme-agnostic. It does not query the watermarked LLM and does not assume any knowledge of the underlying watermarking scheme. Theorem 2 shows that even a small reduction in the average green-token sampling probability over the sequence is sufficient to make the text statistically undetectable. Motivated by this insight, instead of accurately identifying the green tokens as in [1,2], BIRA constructs a proxy target set using token self-information and applies a negative logit bias to these tokens during rewriting. BIRA is also significantly more lightweight than [1,2] (BIRA takes on average 7.95 seconds per sample; see Table 4). Under this more practical threat model, BIRA achieves high evasion rates while preserving semantic fidelity across seven recent watermarking methods.
>
> > Q1. Is the proposed method superior than [1,2]?
> #### Our method achieves over 99% evasion across seven recent watermarking methods, while operating under a more realistic threat model than [1,2] and with substantially lower computational cost, since it does not require issuing a large number of queries to the watermarked LLM.
>
> #### [1] Jovanovic et al. Watermark Stealing in Large Language Models. ICML 2024.
> #### [2] Chen et al. De-mark: Watermark Removal in Large Language Models. ICML 2025.

---

> > ### Comment · Reviewer_PeB8 · 2025-11-25
> > **Official Comment by Reviewer PeB8**
> >
> > Thanks the authors for the rebuttal. However, after carefully checking the rebuttal and reading again the paper, my initial comments are not addressed and there are additional more concerns regarding this paper.
> >
> > > We respectfully clarify that our method differs from [1,2] in both its threat model and methodology, as discussed in the Related Work section. First, we operate under a more realistic threat model.
> >
> > I didn't see the threat model is more realistic compared to [1,2]. Please check the new concerns part below.
> >
> > > Second, our attack is directly motivated by a theoretical characterization of a fundamental vulnerability in current watermarking schemes (Theorem 2).
> >
> > Please note that related work e.g. KGW, DiPmark, have already discussed that the detectability is exponentially proportional to the green list tokens. Theorem 1 and 2 are not new results.
> >
> > > Specifically, [1,2] are query-based attacks that use repeated, carefully designed prompts to identify the green token from the watermarked LLM. This is computationally expensive and has limited practicality in rate-limited API settings.
> >
> > Agree.
> >
> > > Moreover, both methods are specifically tailored to KGW watermarks and rely on specific assumptions:
> >
> > Not true, similarly to BIRA, both methods can work on green-list based watermarks.
> >
> > > [1] assumes knowledge of the hash function that defines the green list, while [2] requires access to the watermarked LLM’s top-$k$ token probabilities, which is typically infeasible for proprietary black-box systems.
> >
> > Not true, both [1,2] worked on black-box scenario.
> >
> > > By contrast, our method (BIRA) is query-free and scheme-agnostic. It does not query the watermarked LLM and does not assume any knowledge of the watermarking scheme.
> >
> > BIRA only work on the green-list based watermarking scheme, thus it does assume knowledge of the watermarking scheme.
> >
> > > Theorem 2 shows that even a small reduction in the green-token sampling probability, on average over the sequence, is sufficient to make the text statistically undetectable.
> >
> > Yes but this is a known results.
> >
> > > Motivated by this insight, instead of accurately identifying the green token as in [1,2], BIRA approximates a proxy green set using token self-information and applies a negative logit bias to these tokens during rewriting. Under this more practical threat model, BIRA achieves high evasion rates while preserving semantic fidelity across seven recent watermarking methods.
> >
> > After carefully checking the paper, I have some **new concerns** regarding the proposed threat model and the method.
> >
> > 1) **High Entropy != Low Probability**. The method relies on the intuition stated in Section 4.2: “Watermarking schemes typically embed their signal in high-entropy (low-probability) tokens to preserve text quality.” This is conceptually inaccurate. Entropy is a property of a **probability distribution**, not of **individual tokens**. A high-entropy distribution simply means probability mass is more evenly spread; it does not imply that tokens necessarily have low probability. In watermarking, signal embedding is easier in high-entropy contexts, because generation is less deterministic. Thus, watermarks exploit high-entropy distributions, not inherently low-probability tokens.
> >
> > 2) **The proxy green list is not reliable**. The method uses a separate reference LM to identify “low-probability” tokens of the original LM. However, the two distributions can differ substantially: the watermarked LM conditions on the full prompt, whereas the reference LM does not. As a result, the proxy green-list may consist of essentially random tokens rather than accurately capturing the low-probability region of the original model’s distribution.
> >
> > Given points (1) and (2), the proposed approach reduces the probability of a largely arbitrary subset of tokens. Consequently, its behavior may resemble that of general paraphrasing attacks rather than a principled or targeted watermark-removal strategy.
> >
> > > Our method achieves over 99% evasion across seven recent watermarking methods, while operating under a more realistic threat model than [1,2] and with substantially lower computational cost, since it does not require issuing a large number of queries to the watermarked LLM.
> >
> > Please add experimental evidence to confirm the superiority of BIRA compared to [1,2].

---

> ### Author Response · Authors · 2025-11-25
> **Response to Reviewer PeB8 (Part 1/2)**
>
> #### We sincerely appreciate your valuable feedback. We have carefully considered your comments and provide detailed responses below.
>
> > We respectfully clarify that our method differs from [1,2] in both its threat model and methodology, as discussed in the Related Work section. First, we operate under a more realistic threat model.
> I didn't see the threat model is more realistic compared to [1,2]. Please check the new concerns part below.
> > [1] assumes knowledge of the hash function that defines the green list, while [2] requires access to the watermarked LLM’s top- token probabilities, which is typically infeasible for proprietary black-box systems.
> Not true, both [1,2] worked on black-box scenario.
>
>
> #### Please refer to the first page, second column of [2], which explicitly states the limitations of [1]: “requiring knowledge of the underlying hash function of the n-gram.” In addition, Appendix A (Limitations) of [2] clearly notes that “the effectiveness of DE-MARK is contingent upon accessing the top-k token probabilities.” These limitations indicate that [1,2] are not a fully black-box scenario.
> > BIRA only work on the green-list based watermarking scheme, thus it does assume knowledge of the watermarking scheme.
>
> #### Thank you for raising this point. While our current theorem is primarily designed for token-level watermarking schemes, our experimental results show that our method is effective against DIP[3], EXP[4],  (distribution-free watermarking). Additionally, we empirically demonstrate that BIRA is effective at evading the state-of-the-art semantic, sentence-level watermark SimMark [5] (Table 7). Although its evasion rate is slightly lower than in the token-level setting, BIRA still achieves strong evasion performance while preserving semantic fidelity. These results show that it remains effective against non-token-based watermarking methods. We will explicitly clarify this point in the revised manuscript.
>
>
> > Second, our attack is directly motivated by a theoretical characterization of a fundamental vulnerability in current watermarking schemes (Theorem 2).
> Please note that related work e.g. KGW, DiPmark, have already discussed that the detectability is exponentially proportional to the green list tokens. Theorem 1 and 2 are not new results.
> #### Regarding the novelty of Theorems 1 and 2, we respectfully clarify that our contribution lies in the adversarial formulation of the problem, which fundamentally differs from prior works such as KGW and DiPmark in two key ways:
> - #### Conceptual shift (robustness → vulnerability). Prior work (e.g., KGW, DiPmark) uses concentration inequalities to derive robustness guarantees, i.e., conditions under which detection is guaranteed to succeed, and analyzes the detector under **IID Bernoulli green indicators** that derive **bounds of the detection probability in terms of the number of green tokens**. By contrast, we recast the problem in terms of vulnerability: Theorem 2 gives an upper bound on detection probability when an attacker suppresses the average green-token probability by a margin $\delta$. Specifically, our theorem assumes an **arbitrary (non-iid) attack generator** and provides a **condition on the sequence of conditional probabilities of sampling a green token** under which the detection probability decays exponentially. This attack-oriented, detector-agnostic formulation provides a theoretical basis for minimizing the watermark signal rather than maximizing its retention, and it directly motivates the design of BIRA.
> - #### Continuous vs. discrete perturbations. DiPmark’s analysis of robustness focuses on discrete text edits (insertions, deletions, substitutions), deriving a certified radius in terms of the maximum number of token edits that preserve detectability. In contrast, Theorem 2 models vulnerability under continuous logit suppression. We analyze the effect of applying a global negative bias to a proxy green set, which effectively reduces the green-token probability. This distinction directly motivates the design of BIRA, which applies a negative logit bias instead of performing discrete edits and thereby achieves high watermark evasion rates.

---

> ### Author Response · Authors · 2025-11-25
> **Response to Reviewer PeB8 (Part 2/2)**
>
> > Motivated by this insight, instead of accurately identifying the green token as in [1,2], BIRA approximates a proxy green set using token self-information and applies a negative logit bias to these tokens during rewriting. Under this more practical threat model, BIRA achieves high evasion rates while preserving semantic fidelity across seven recent watermarking methods.
> After carefully checking the paper, I have some new concerns regarding the proposed threat model and the method.
> > High Entropy != Low Probability. The method relies on the intuition stated in Section 4.2: “Watermarking schemes typically embed their signal in high-entropy (low-probability) tokens to preserve text quality.” This is conceptually inaccurate. Entropy is a property of a probability distribution, not of individual tokens. A high-entropy distribution simply means probability mass is more evenly spread; it does not imply that tokens necessarily have low probability. In watermarking, signal embedding is easier in high-entropy contexts, because generation is less deterministic. Thus, watermarks exploit high-entropy distributions, not inherently low-probability tokens.
>
>
> #### Thank you for raising this point. We agree that the wording “high-entropy (low-probability) tokens” in Section 4.2 is conceptually inaccurate, since entropy is a property of the full distribution rather than individual tokens. In our method, we actually use token-level self-information (surprisal) to select low-probability tokens. We will explicitly revise the manuscript to clarify this.
>
>
> > The proxy green list is not reliable. The method uses a separate reference LM to identify “low-probability” tokens of the original LM. However, the two distributions can differ substantially: the watermarked LM conditions on the full prompt, whereas the reference LM does not. As a result, the proxy green-list may consist of essentially random tokens rather than accurately capturing the low-probability region of the original model’s distribution.
> Given points (1) and (2), the proposed approach reduces the probability of a largely arbitrary subset of tokens. Consequently, its behavior may resemble that of general paraphrasing attacks rather than a principled or targeted watermark-removal strategy.
>
>
> #### While the reference LM is different from the watermarked LM, our attack does not rely on accurately recovering the true green set as in [1,2]. By Theorem 2, it is sufficient to slightly reduce the average green-token sampling probability over the sequence, so exact identification of green tokens is not required. In Appendix A.1, we further analyze robustness to proxy green sets and show that the attack remains effective even when the proxy green set is not equal to the true green set. Additionally, Appendix C provides a detection-bound analysis demonstrating that BIRA substantially lowers the detection probability than vanilla paraphrasing. Empirically, we validate the effectiveness of our method across three different LMs (Llama-3.1-8B,70B, GPT-4o-mini), and Figure 5 shows that self-information–guided selection consistently outperforms random selection, indicating that it more effectively targets green tokens. Since watermarked text is dominated by green tokens, even random selection naturally captures many of them. Therefore, the additional performance gain from our method confirms that high self-information is significantly more correlated with the watermark signal than random token selection. Finally, BIRA also significantly outperforms vanilla paraphrasing (Table 1), which further supports that it is a targeted and effective watermark-removal strategy rather than a general rewriting attack.
> > Moreover, both methods are specifically tailored to KGW watermarks and rely on specific assumptions:
> Not true, similarly to BIRA, both methods can work on green-list based watermarks.
> >Please add experimental evidence to confirm the superiority of BIRA compared to [1,2].
>
>
> #### The evaluation in [1,2] primarily focuses on KGW watermarks and does not include more recent semantic watermarking schemes such as SIR, and distribution-free watermarking such as DIP [3] and EXP [4]. In contrast, our experiments demonstrate that BIRA is effective across seven watermarking methods, including SIR, DIP, and EXP. We are currently implementing the attacks from [1,2] against green-list-based watermarks, and we will report these additional results in the updated manuscript.
>
>
> #### [1] Jovanovic et al. Watermark Stealing in Large Language Models. ICML 2024.
> #### [2] Chen et al. De-mark: Watermark Removal in Large Language Models. ICML 2025.
> #### [3] Wu et al. A resilient and accessible distribution-preserving watermark for large language models. ICML 2024.
> #### [4] Aaronson et al. Watermarking gpt outputs. https://www.scottaaronson.com/ talks/watermark.ppt, 2022
> #### [5] A DabiriAghdam et al. SimMark: A Robust Sentence-Level Similarity-Based Watermarking Algorithm for Large Language Models. EMNLP 2025

---

> > ### Comment · Reviewer_PeB8 · 2025-11-26
> > **Official Comment by Reviewer PeB8**
> >
> > After carefully checking the author's comments and the proposed Algorithm 1, the reviewer found some more severe problems.
> >
> > > Thank you for raising this point. We agree that the wording “high-entropy (low-probability) tokens” in Section 4.2 is conceptually inaccurate, since entropy is a property of the full distribution rather than individual tokens. In our method, we actually use token-level self-information (surprisal) to select low-probability tokens. We will explicitly revise the manuscript to clarify this.
> >
> > Thanks for confirming this. Now the question become, **how can the proposed algorithm still work when it starts from an incorrect initial assumption?**
> >
> > > While the reference LM is indeed different from the watermarked LM, our attack does not rely on accurately recovering the true green set as in [1,2]...
> >
> > It is well known that reducing green-list tokens directly harms detectability. If BIRA cannot correctly reconstruct the green list, then its attack strength must come almost entirely from the paraphrasing stage, since paraphrasing alone can already decrease the proportion of green-list tokens. In Algorithm 1, the authors introduce an additional resampling step using a reference model, it appears that most of the attack effect may simply result from this resampling procedure rather than any specialized design.
> >
> > Besides, the authors confirm that no prompt is included in the BIRA algorithm, which makes the proposed algorithm susceptible. In particular, in the resampling stage (lines 8–20 of Algorithm 1), the process surprisingly begins from an empty sequence (line 9) with no conditioning prompt. **It is unclear how BIRA could meaningfully reproduce the intended semantic content without a prompt.** The constraints introduced in line 17 alone cannot guide a model to generate a coherent, unwatermarked sentence starting from nothing. Without a prompt, generating the target content is essentially impossible.
> >
> >
> > Below are the minor points
> >
> > > In addition, Appendix A (Limitations) of [2] clearly notes that “the effectiveness of DE-MARK is contingent upon accessing the top-k token probabilities.” These limitations indicate that [1,2] are not a fully black-box scenario.
> >
> > Please check table 2 and 3 in [2], while the authors provided black-box results.
> >
> > > Regarding the novelty of Theorems 1 and 2
> >
> > Please note that BIRA decrease the detectability through reducing the number of the green tokens, which is achieved by "continuously modifing the green token logits". Since the number of the green token is discrete, the attack is also an example of discrete attack.
> >
> > > The evaluation in [1,2] primarily focuses on KGW watermarks and does not include more recent semantic watermarking schemes such as SIR, and distribution-free watermarking such as DIP [3] and EXP [4]. In contrast, our experiments demonstrate that BIRA is effective across seven watermarking methods, including SIR, DIP, and EXP. We are currently implementing the attacks from [1,2] against green-list-based watermarks, and we will report these additional results in the updated manuscript.
> >
> > In Appendix 4 of [2], the authors provide experimental result on DIP.

---

> > > ### Author Response · Authors · 2025-11-26
> > > **Response to Reviewer PeB8 (Part 1/2)**
> > >
> > > #### We sincerely appreciate your prompt response and active engagement. We carefully provide our detailed responses below.
> > > > While the reference LM is indeed different from the watermarked LM, our attack does not rely on accurately recovering the true green set as in [1,2]...
> > >
> > >
> > > #### For clarity, Theorem 2 shows that an approximate green list is sufficient for the attack; the exact true green set is not necessary. It establishes that reducing the average green-token sampling probability by a small margin ($\delta$) across the sequence is enough to make detection statistically unlikely, as the detection probability then decays exponentially. Thus, we only need a proxy set that is sufficiently correlated with the true green set to suppress the average green-token rate. The effectiveness of this token selection strategy is detailed below.
> > >
> > > >  It is well known that reducing green-list tokens directly harms detectability. If BIRA cannot correctly reconstruct the green list, then its attack strength must come almost entirely from the paraphrasing stage,
> > > > Its attack strength must come almost entirely from the paraphrasing stage. Since paraphrasing alone can already decrease the proportion of green-list tokens.
> > > >  In Algorithm 1, the authors introduce an additional resampling step using a reference model, it appears that most of the attack effect may simply result from this resampling procedure rather than any specialized design.
> > >
> > >
> > > #### The vanilla paraphrasing alone can not decrease the proportion of green-list tokens as its attack results are substantially lower than those of BIRA with negative logit bias (see Tables 1, especially for the SIR watermark). In particular, for GPT-4o-mini, BIRA achieves significantly higher ASR than vanilla paraphrasing under the same settings, indicating that the attack strength does not entirely come from paraphrasing alone.
> > > #### Moreover, our method does not perform any additional resampling step. BIRA simply applies a negative logit bias to the proxy green tokens during each token generation step, using the same paraphrasing system prompt (Appendix I) with a given watermarked text as all baselines for a fair comparison.
> > > #### Additionally, Figure 5 shows that self-information–guided selection consistently outperforms random selection, indicating that it more effectively targets green tokens. Since watermarked text is dominated by green tokens, even random selection will hit many of them. The additional performance gain of BIRA therefore confirms that high self-information is significantly more correlated with the watermark signal than random token selection, which shows that our attack is specifically tailored to watermark evasion.
> > >
> > > > Besides, the authors confirm that no prompt is included in the BIRA algorithm, which makes the proposed algorithm susceptible. In particular, in the resampling stage (lines 8–20 of Algorithm 1), the process surprisingly begins from an empty sequence (line 9) with no conditioning prompt. It is unclear how BIRA could meaningfully reproduce the intended semantic content without a prompt. The constraints introduced in line 17 alone cannot guide a model to generate a coherent, unwatermarked sentence starting from nothing. Without a prompt, generating the target content is essentially impossible.
> > >
> > >
> > > #### Thank you for raising this point. For ease of presentation, we omitted the full input: the system paraphrasing prompt (Appendix I) together with the given watermarked document. In Algorithm 1, the “empty sequence” simply denotes the buffer that will be filled with newly generated tokens. We will explicitly include this prompt conditioning and clarify this initialization step in Algorithm 1 in the revised manuscript to avoid confusion.

---

> ### Author Response · Authors · 2025-11-26
> **Response to Reviewer PeB8 (Part 2/2)**
>
> > In addition, Appendix A (Limitations) of [2] clearly notes that “the effectiveness of DE-MARK is contingent upon accessing the top-k token probabilities.” These limitations indicate that [1,2] are not a fully black-box scenario.
> Please check table 2 and 3 in [2], while the authors provided black-box results.
>
>
> #### Our work specifically focuses on **watermark removal attacks**. In [2], **Section 3.1 Threat models for “Watermark Removal”** explicitly states that “We consider a **gray-box setting**, where the attacker is granted access to the top-k probabilities of tokens generated by the model. … We did not consider black-box settings in our study because without token probabilities, it is not possible to theoretically bound the gap between the original and post-removal distributions of the LM.”
>
>
> #### Thus, for watermark removal attack, [2] assumes a gray-box setting with access to top-k token probabilities, consistent with the limitation noted in Appendix A (“a requirement that may not always be feasible in restricted or proprietary environments”). We note that the **black-box scenario in [2] applies only to watermark exploitation (spoofing) attacks**. In contrast, BIRA is designed for a fully black-box, query-free removal scenario that does not require access to top-k token probabilities of the watermarked LLM, which is a more realistic threat model than [2].
>
>
> > Regarding the novelty of Theorems 1 and 2. Please note that BIRA decrease the detectability through reducing the number of the green tokens, which is achieved by "continuously modifing the green token logits". Since the number of the green token is discrete, the attack is also an example of discrete attack.
>
> #### Our distinction between “continuous” and “discrete” was not about the discreteness of the **output sequence**, but about the **space in which the attack is defined and analyzed**. Our theorem and algorithm operate in logit/probability space, where the control variable (the negative bias) is continuous. BIRA perturbs the distribution before sampling by modifying logits, rather than imposing or counting discrete token edits after generation, as in DiPMark’s edit-based robustness analysis. To avoid confusion, we will revise the manuscript to describe this distinction.
>
>
> > In Appendix 4 of [2], the authors provide experimental result on DIP.
> #### Thank you for pointing this out. We are aware that Appendix 4 of [2] reports watermark-removal results on DiPMark under a gray-box setting. To further validate the effectiveness of our method, we are currently running the experiment for the removal attack from [2].

---

> ### Comment · Reviewer_PeB8 · 2025-11-26
> **Official Comment by Reviewer PeB8**
>
> Before proceeding with further discussion, I think we should first clarify one important point: **the recovered token list is not an “approximate” green-list**. There are two reasons for this:
>
> 1) There is no evidence that the low-probability tokens identified from the recovered distribution correspond to the actual green tokens.
>
> 2) The recovered LM distribution can differ substantially from the original LM distribution, especially when the prompt used to generate the watermarked sequence is unavailable.
>
> For these reasons, **I strongly suggest that the authors stop from referring to the recovered token list as an “approximate” green-list**, as this terminology is misleading.
>
> In fact, in KGW the green-list tokens are randomly selected from the vocabulary in every generation step, and each token has an equal chance of being assigned to either the red list or the green list. The green list does not encode any preference based on token probability—both high-probability and low-probability tokens are equally likely to be included.

---

> ### Author Response · Authors · 2025-11-26
> **Response to Reviewer PeB8**
>
> #### Thank you for this clarification and for pointing out the potential confusion in our terminology. As shown in [1], targeting high-self-information tokens is an effective way to suppress watermark signals. That work theoretically analyzes how watermarking affects token self-information (Appendix G and H) and empirically shows that self-information–based masking and rewriting can effectively remove the watermark signal, albeit with a substantial cost in text quality. We discuss the differences between our method and SIRA [1] in the Remark in Section 4.2.
>
> #### We revised the manuscript to include both the system prompt and the target watermarked text in Algorithm 1 of our method. We also clarified that we use token self-information rather than the term token entropy.
>
> #### [1] Cheng et al., “Revealing Weaknesses in Text Watermarking Through Self-Information Rewrite Attacks.” ICML 2025

---

> ### Author Response · Authors · 2025-11-26
> **Response to Reviewer PeB8 on the Comparison with DeMark**
>
> #### Thank you for taking the time to review and engage with our paper. We provide our response regarding the comparison with DeMark below.
>
> > Our method achieves over 99% evasion across seven recent watermarking methods, while operating under a more realistic threat model than [1,2] and with substantially lower computational cost, since it does not require issuing a large number of queries to the watermarked LLM.
>
> > Please add experimental evidence to confirm the superiority of BIRA compared to [1,2].
> #### The goal of our work and that of [2] is fundamentally different. De-Mark [2] treats the **model as the attack target**: it first steals watermark parameters and then adjusts the watermarked LLM’s probabilities so its behavior approximates the original, unwatermarked model. Our setting instead treats the **watermarked text as the target**: given a watermarked text, we rewrite it via BIRA to remove the watermark signal. Since one operates in model space and the other in text space, a direct comparison is not well defined.
> #### Moreover, [2] assumes access to the watermarked LLM’s top-k token probabilities and the ability to modify its logits, and in our runs, its removal attack takes around **one hour per sample** under their default setup. Such probability access and computational cost are unrealistic for many commercial LLMs (e.g., Gemini with SynthID watermark), whereas BIRA operates in a fully black-box setting and is computationally lightweight (BIRA takes on average **7.95 seconds** per sample; see Table 4).
>
> #### [2] Chen et al. De-mark: Watermark Removal in Large Language Models. ICML 2025.

---

### Official Review · Reviewer_ELUQ · 2025-11-01

**Soundness:** 3
**Presentation:** 3
**Contribution:** 2
**Rating:** 4
**Confidence:** 4

**Summary:**

The paper studied the robustness of watermarking in adversarial attack settings. To address the problem and understand it deeply, the authors introduced a method called the bias-inversion rewrite attack, whose application is agnostic to specific models. The experiments show that the method can achieve a high rate of evasion while preserving the original semantic content.

**Strengths:**

1, The problem setting is clearly presented;

2, The method seems to address the problem well from the experiments

**Weaknesses:**

The rewriting attack is common in recent literature. It would be great if the authors could show how their algorithm differs from others. The experiments also show that their method is indeed effective. It would also be great to link their theoretical analysis with the experiments.

**Questions:**

Seems that only 1 dataset is involved in your experiments. Will it be more convincing to conduct experiments on much more datasets?

---

> ### Author Response · Authors · 2025-11-20
> **Response to Reviewer ELUQ**
>
> #### Thank you for taking the time to review and engage with our paper. We have carefully considered your comments and provide detailed responses below. We have also added the corresponding experimental results and analyses to Appendix C and Appendix D in the revised manuscript, marked in pink.
>
> > W1. The rewriting attack is common in recent literature. It would be great if the authors could show how their algorithm differs from others.
>
> #### While paraphrasing attacks [1,2,3] are common in recent literature, our method differs in both threat model and methodology. First, we operate under a more realistic threat model. Second, our attack is directly motivated by a theoretical characterization of a fundamental vulnerability in current watermarking schemes (Theorem 2).
>
> #### Specifically, existing attacks rely on assumptions that are often impractical in real-world settings:
> - #### [1,2] are query-based attacks that use repeated, carefully designed prompts to recover green tokens from the watermarked LLM. This is computationally expensive and has limited practicality in rate-limited API settings. Moreover, both methods are tailored to KGW watermarks and rely on specific assumptions: [1] assumes knowledge of the hash function that defines the green list, while [2] requires access to the model’s top-$k$ token probabilities, which is typically infeasible for proprietary black-box systems.
> - #### [3] is a query-free attack but assumes knowledge of the watermarking scheme and considers an adaptive scenario in which the adversary fine-tunes a specialized paraphrasing model for the target watermark, relying on transferability to unseen schemes.
>
> #### By contrast, our method (BIRA) is **query-free, scheme-agnostic, and training-free**. It does not query the watermarked LLM, does not assume any knowledge of the watermarking scheme, and does not require training an additional paraphrasing model. Theorem 2 shows that even a small reduction in the green-token sampling probability, on average over the sequence, is sufficient to make the text statistically undetectable. Motivated by this insight, instead of accurately identifying the green list as in [1,2] or training a specialized paraphrasing model as in [3], BIRA approximates a proxy green set using token self-information and applies a negative logit bias to these tokens during rewriting. Under this more practical threat model, BIRA achieves high evasion rates (>99%) while preserving semantic fidelity across seven recent watermarking methods.
>
> > W2. It would also be great to link their theoretical analysis with the experiments.
>
> #### To validate Theorem 2, in Appendix C, we compute the upper bound on the detection probability for each sample under the Unigram watermark using the experimental setup described in Section 5.1. Figure 6 shows that BIRA achieves significantly lower per-sample detection bounds than Vanilla for most samples, indicating a lower detection probability and consequently more effective watermark evasion.
>
> > Q1. Seems that only 1 dataset is involved in your experiments. Will it be more convincing to conduct experiments on much more datasets?
>
> #### We conducted additional experiments on two datasets, Dolly CW and the DBPEDIA Class dataset, using 100 examples and the experimental setup described in Section 5.1 with Llama-3.1-8B. The experimental results in Appendix D show that our method outperforms the baselines in both attack efficacy and text quality, consistent with the main results.
>
> #### [1] Jovanovic et al. Watermark Stealing in Large Language Models. ICML 2024.
> #### [2] Chen et al. De-mark: Watermark Removal in Large Language Models. ICML 2025.
> #### [3] Diaa et al. Optimizing Adaptive Attacks against Watermarks for Language Models. ICLR 2025 workshop

---

> ### Author Response · Authors · 2025-11-27
> **A Gentle Reminder**
>
> #### We would like to kindly remind you that the discussion period will end in 6 days. We would greatly appreciate it if you could let us know whether our rebuttal has sufficiently addressed your concerns. Thank you once again for your time and effort.

---

### Meta-Review · Area_Chair_RYGY · 2025-12-22

**Summary:**

LLM watermarks are crucial for tracing AIGCs, yet several attacks have demonstrated their poor robustness. This paper proposes a novel evasion attack named Bias-Inversion Rewriting Attack (BIRA). The core idea of BIRA is to suppress the logits of likely watermarked tokens during LLM-based rewriting. However, the Area Chair still has the following concerns after checking all the reviews and discussions.

- Although experiments show that BIRA can achieve strong watermark evasion effects, the underlying mechanism remains heuristic. There is no sufficient theoretical analysis to prove that the recovered token list is an “approximate” green list. The AC also questions the fundamental feasibility of recovering the green list, as pointed out by the reviewer PeB8, each token has an equal chance of being assigned to either the red list or the green list in the KGW scheme.

- Experiments use a proxy model identical to the target model (besides attacking against GPT-4o-mini) to generate the proxy green list, which fails to demonstrate the generalization of BIRA and even contradicts the assumed threat model (e.g., " the adversary has no knowledge of the target model").

- Given that existing evasion attacks already achieve high success rates, and this paper does not propose any potential defenses, its contribution remains incremental.

**Reviewer Concerns:**

Addressed Concerns:
- Clarify differences in attacking settings from existing attacks.
- Add attack performance against sentence-level watermarking schemes.
- Add attack performance on more datasets.


Unaddressed Concerns:
- Fail to prove the recovered token list is an “approximate” green-list.
- Lack of potential defenses.

**Reviewer Scores:**

I believe that the reviewers have engaged in thorough discussion, and the current scores are reasonable. However, some ratings remain on the higher side, possibly due to insufficient consideration of the comments provided by other reviewers.

---

### Decision · Program_Chairs · 2026-01-26

Reject